

# Description and applications of a mobile system performing on-road aerosol remote sensing and in situ measurements

Ioana Elisabeta Popovici[1,2], Philippe Goloub[1], Thierry Podvin[1], Luc Blarel[1], Rodrigue Loisil[1], Florin Unga[1], Augustin Mortier[3], Christine Deroo[1], Stéphane Victori[2], Fabrice Ducos[1], Benjamin Torres[4,1], Cyril Delegove[1], Marie Choël[5], Nathalie Pujol-Söhne[6] and Christophe Pietras[7]

[1]Univ. Lille, CNRS, UMR 8518 – LOA – Laboratoire d'Optique Atmosphérique, F-59000 Lille, France
[2]Cimel Electronique, 75011 Paris, France
[3]Norwegian Meteorological Institute, 0313 Oslo, Norway
[4]GRASP-SAS, Remote sensing developments, Univ. Lille, 59650 Villeneuve d'Ascq, France
[5]Univ. Lille, CNRS, UMR 8516 – LASIR – Laboratoire de Spectrochimie Infrarouge et Raman, F-59000 Lille, France
[6]ATMO Hauts-de-France, 59000 Lille, France
[7]Ecole Polytéchnique, CNRS, Laboratoire de Météorologie Dynamique, 91120 Palaiseau, France

*Correspondence to*: Ioana E. Popovici (ie.popovici@ed.univ-lille1.fr)

**Abstract.**

The majority of ground-based aerosols observations are limited to fixed locations, narrowing the knowledge on their spatial variability. In order to overcome this issue, a compact Mobile Aerosol Monitoring System (MAMS) was developed to explore the aerosol vertical and spatial variability. This mobile laboratory is equipped with a micropulse lidar, a sun-photometer and an aerosol spectrometer. It is distinguished by other transportable platforms through its ability to perform on-road measurements and its unique feature lies in the sun-photometer capable to track the sun during motion. The system presents a great flexibility, being able to respond quickly in case of sudden aerosol events such as pollution episodes, dust, fire or volcano outbreaks. On-road mapping of aerosol physical parameters such as attenuated aerosol backscatter, aerosol optical depth, particle number and mass concentration and size distribution is achieved through the MAMS. The performance of remote sensing instruments on-board has been evaluated through intercomparison with instruments in reference networks (i.e. AERONET and EARLINET), showing that the system is capable of providing high quality data. This also illustrates the application of such system for instrument intercomparison field campaigns. Applications of the mobile system have been exemplified through two case studies in northern France. MODIS AOD data was compared to ground-based mobile sun-photometer data. A good correlation was observed with $R^2$ of 0.76, showing the usefulness of the mobile system for validation of satellite-derived products. The performance of BSC-DREAM8b dust model has been tested by comparison of results from simulations to the lidar-sun-photometer derived extinction coefficient and mass concentration profiles. The comparison indicated that observations and model are in good agreement in describing the vertical variability of dust layers. Moreover, on-road measurements of $PM_{10}$ were compared with modelled $PM_{10}$ concentrations and with ATMO Hauts-de-France and AIRPARIF air quality in situ measurements, presenting an excellent agreement in horizontal




spatial representativity of PM$_{10}$. This proves a possible application of mobile platforms for evaluating the chemistry-models performances.

## 1 Introduction

Aerosols are a variable component of the atmosphere impacting air quality and climate. In order to monitor the atmospheric

aerosols, independent ground-based observations are performed globally, grouped in large networks, such as the Aerosol Robotic Network (AERONET; Holben et al., 1998), the Micropulse Lidar Network (MPLNET; Welton et al., 2005), the EARLINET/ACTRIS (Aerosol, Clouds and Trace gases Research Infrastructure Network; Pappalardo et al., 2014), or organized in national surface networks, dedicated to air quality monitoring (ATMO France, http://www.atmo-france.org/). Such observations offer capabilities for long-term monitoring of aerosol properties and evaluation of trends, but they are

limited over fixed locations. Lidars are excellent tools for studying the height-resolved aerosol characteristics, especially interesting for pollution episodes and long-range transport situations. Such observations of vertical aerosol structures and quantification of their contribution to the total column aerosol content are important as the lifetime of particles in the free troposphere is of the order of weeks. Furthermore, concentrations at surface level can sometimes be influenced by the subsidence of transported aerosols from the free troposphere. Thus, the knowledge of the vertical distribution, as close to the

surface as possible, of aerosols optical properties, is required to precisely predict aerosol concentrations, especially for air quality models. Gravimetric measurements of particles mass concentration at surface level are sparse within the territory, which directs us to the alternative of using optical aerosol properties to estimate their vertically-resolved mass concentration and to get as accurate estimates as possible for particle concentration at surface level. This direction is a challenge, as it requires information on aerosol chemical and microphysical properties. Nevertheless, in this work we exemplify this

direction on mobile measurements.

Existing lidar networks consist of systems with various configurations, from single-wavelength elastic to multi-wavelength Raman lidars. Most of them are complex instruments that require regular maintenance and controlled environment for their operation, so they are predominantly installed in laboratory rooms. Therefore, their use for atmospheric profiling is limited over a fixed location. Nevertheless, the aerosol distribution is highly variable spatially in case of sudden events; e.g.

pollution episodes, dust and fire outbreaks, volcano eruptions, long-range transports and dispersion of pollutants from emission sources. In these situations, field observations are important, as the spatial variability is impossible to assess from point measurements. Mobile observations are one of the best solutions to map the extent of such events and to study the regional gradients in aerosol concentrations. If lidars could be easily deployed at the time needed, the number of applications would rapidly increase, from the validation of satellite measurements and model predictions to the investigation of pollutants

dynamics and quantification of diffuse emissions at industrial sites.

A number of mobile on-road experiments focused on the spatial variability of the particle number and mass concentration along highways in Jordan (Hussein et al., 2017), in Noord, Holland (Weijers et al., 2004) and in an Alpine Valley (Weimer



et al., 2009), of black carbon and particulate sulphates concentrations from Mainz, Germany to southern Spain (Drewnick et al., 2012) and of aerosol and gas phase ambient concentrations in Zurich, Switzerland (Bukowiecki et al., 2002). Other studies investigated the tropospheric trace gas distribution: $NO_2$ along Brussels-Heidelberg transect (Wagner et al., 2010), $SO_2$ and $NO_2$ in Guangzhou, China (Wu et al., 2013) and $CO_2$ and $CH_4$ in Utah, USA (Bush et al., 2015). Remote sensing

instruments are nonetheless less involved in mobile applications due to their size, to operation cautions and sensitivity to misalignment due to movement. Mobile sun photometer measurements on ship cruises are performed with Microtops II handheld sun photometers in the framework of the Maritime Aerosol Network (MAN; Smirnov et al., 2009), a component of AERONET, dedicated to collect aerosol optical depth data over oceans. In the early developments of our mobile system, a Microtops II sun photometer was used. Other mobile sun photometer measurements have been reported by Lewandowski et

al. (2010), but they refer to stationary measurements at different points along the road, the sun photometer being installed on the roof of the vehicle for the measurements and taken down during travel. The first evidence of continuous mobile sun photometer measurements during vehicle's movement has been presented by Mortier et al. (2012) during the DRAGON (Distributed Regional Aerosol Gridded Observation Networks; Holben et al., n.d.) – USA campaign. For lidars, the term "mobile" refers mostly to scanning (Chiang et al., 2015; Marchant, 2009), transportable (Berkhout et al., 2016; Chazette et

al., 2014; Freudenthaler et al., 2002) or scanning and transportable (Dou et al., 2014) systems, used for measurements in remote places, during field campaigns or simply outside laboratory rooms. To our knowledge, only few studies presenting ground-based lidar profiling of aerosol properties by on-road mobile measurements have been conducted, over Paris agglomeration (Pal et al., 2012; Raut and Chazette, 2009; Royer et al., 2011), on London's orbital motorway (Raut et al., 2009) and from Paris to Siberia (Dieudonné et al., 2015). Thus, we consider that the capability of deploying remote sensing

instruments for on-road mapping of aerosol properties is not enough explored. For the first time, the description of a compact mobile system performing on-road measurements with a lidar, a sun photometer and an aerosol spectrometer, is presented here. The uniqueness of the system lies mainly in the sun photometer, capable to track the sun during vehicle's motion, and in the synergy of lidar and sun photometer measurements to derive aerosol optical properties profiles along the investigated route.

The main objective of this paper is to show the potential of a mobile platform performing on-road remote sensing and in situ measurements to derive aerosol properties. The use of the mobile system for atmospheric studies is versatile, some of its possible applications being emphasized throughout this study. Section 2 presents the mobile system and the instrumentation integrated in the payload. Section 3 describes the data processing and quality assurance procedures, the methodology used to derive aerosol properties from remote sensing measurements and a discussion on uncertainties. Furthermore,

intercomparisons with reference instruments are performed and are included in Sect. 3. In Sect. 4 two case studies using data from mobile campaigns in northern France and Paris are presented in Sect. 4, examples that illustrate the capability of our system to evaluate the mass concentration both vertically-resolved and at surface level. In addition, at surface level, the lidar-derived mass concentrations are compared to air quality stations measurements and to modelled $PM_{10}$ concentrations. The last section (Sect. 5) is dedicated to conclusions.



## 2 Description of the mobile system and instrumentation

Most instruments set up on mobile platforms and deployed in field campaigns are bulky systems, requiring large vehicles for their installation and transportation. As compared to such systems, the MAMS (*Mobile Aerosol Monitoring System*) is more compact and robust. The mobile laboratory described here is a minivan equipped with a micro-pulse lidar, a sun photometer and an aerosol spectrometer (Fig. 1). The minivan is a Renault Kangoo Intens Energy 115CH CO2 140gr/km (length: 4.21 m, width: 1.83 m, total height: 1.87 m). A car with gasoline engine was chosen against one on diesel due to lower particle emissions and against an electric car due to higher autonomy (around 500 km). The vehicle's trunk and rear part of the roof have been modified to allow the installation of instruments inside and on top of the vehicle. In order to minimize shocks and vibrations on the instruments while driving, the rear wheels are equipped with coil suspension and instruments are equipped with shock-absorbing devices. For lidar measurements, a 350 x 250 mm anti-reflective glass of 8 mm thickness type Conduran magic, that has a transmission > 90% for 532 nm, has been embedded in the rear-part of the car's roof. In order to continuously operate the instruments, a 12V/300Ah AGM battery and a sine-wave inverter-charger are mounted in the vehicle. With a power consumption of 100 W, an autonomy of 29 hours can be ensured for continuous measurements. The vehicle is also equipped with an electrical installation and an external outlet used for charging the battery. During stationary measurements, the battery is fed and recharged using the external outlet. The total payload including measurement equipment, battery and inverter is approximately 130 kg.

The lidar included in the MAMS is the ***CE370 micro-Lidar*** (Pelon et al., 2008) designed to monitor aerosols and clouds in the troposphere, typically up to 15 km with a vertical resolution of 15 m. Maximum ranges between 15 km and 20 km can be reached for moderate aerosol loadings. It is a one-channel elastic backscattering lidar operating at 532 nm with 20 µJ pulse energy and is developed and commercialized by CIMEL Electronique (www.cimel.fr). The instrument's design consists of a shared transmitter-receiver telescope (mono-axis system) connected to the control and acquisition unit through a 10 m optical fiber. The advantage of the 10 m optical fiber is that it allows deploying the lidar outside of the vehicle for static angular measurements for example. The lidar is autonomous, lightweight and compact, it fulfils eye-safety standards and requires no special authorization for its operation. These features make it suitable for a mobile system and for continuous, non-supervised operation. The CE370 micro-lidar emits laser pulses, at a 4.7 kHz repetition rate, which can be accumulated over 30 seconds or 1 minute. For a typical vehicle speed of 110 km/h this corresponds to a spatial horizontal resolution of approximately 900 m and 1800 m respectively. The lidar control and acquisition unit, enclosing the optical and electronic components, is installed on vibration isolators to minimize shocks while the vehicle is moving. The lidar transmitter-receiver telescope is fixed to the van's floor with a mechanical support, in order to ensure stability during movement. Furthermore, the shared telescope design eliminates any optical misalignment of the emission and reception channels. Also, an inclinometer is attached to the optical head to correct for the platform inclination when driving on steep slopes or for the case when static angular measurements are performed. For the mobile measurements, lidar sounding is performed only in the





zenith direction. A second lidar (model CE376 GPNP, CIMEL Electronique) with three acquisition channels, two at 532 nm (elastic and perpendicularly polarized backscattering) and one at 808 nm (elastic backscattering), is planned to be integrated in the mobile system's payload after testing and validation with fixed observations.

The *sun-tracking photometer PLASMA* (Photomètre Léger Aéroporté pour la Surveillance des Masses d'Air), developed by Laboratoire d'Optique Atmospherique (LOA), is set up on the roof of the mobile system for continuous measurements of direct spectral solar radiation. Spectral extinction Aerosol Optical Depths (AOD) are derived from PLASMA measurements. The sun photometer has 9 channels at similar spectral range typical for standardized network instruments: 0.339 μm, 0.379 μm, 0.440 μm, 0.500 μm, 0.674 μm, 0.870 μm, 1.019 μm and 1.643 μm and 0.94 μm for the atmospheric precipitable water

vapor content. The single-band filters inserted in the optical path are distributed into two motor-driven filter wheels, one for the visible and near-infrared wavelengths and one for the infrared short-wavelengths. The second wheel contains only the 1.643 μm channel so it does not turn for the current configuration, but more filters can be added. The full angle field of view (FOV) of the two collimators is 1.5°, which is comparable to the 1.2° FOV of CIMEL photometers, limiting sensitivity to atmospheric scattering. PLASMA can move in elevation (0-88°) and azimuth (0-360°) and can rotate in azimuth thanks to a

ring power connector, which makes it suitable for mobile applications. A GPS system delivers geographical coordinates used to determine the position of the Sun. The Sun tracking is achieved with the use a four-quadrant detector and a compass giving the heading, pitch and roll of the mobile platform. A more detailed description of the instrument and application to airborne measurements is presented by Karol et al. (2013). PLASMA sun photometer has been also used for airborne measurements up to 12 km on the French Falcon 20 aircraft during the AEROCLO-SA field campaign over the South

Atlantic Ocean as well as during the SHADOW field campaign in M'Bour, Senegal, on-board an ultralight (ULM) aircraft. The acquisition time for a complete sequence of direct-sun radiance measurements for all filters is approximately 1.9 seconds. As data are filtered for clouds and other obstacles encountered in the line of sight of the sun photometer during motion (trees, buildings, bridges etc.), the temporal sampling is decreased to approximately 10 seconds. For a typical vehicle speed of 110 km/h this corresponds to a spatial horizontal resolution of approximately 300 m. The cloud screening is

achieved using the triplet variability criteria applied for any standard sun photometer in AERONET network (Smirnov et al., 2000).

A portable *aerosol spectrometer* (model Mini-WRAS 1371, Grimm Aerosol Technik) is integrated in the MAMS payload for real-time measurements of airborne particle size and mass distribution. Mass concentrations for the $PM_1$, $PM_{2.5}$ and $PM_{10}$

size fractions are determined based on optical size distribution measurements and assuming a density of 1.7 g cm$^{-3}$, typical value for urban particles. The instrument classifies particles by their electrical mobility diameter and optical diameter in 40 size-bins, from 0.01 to 35 μm every minute with a flow rate of 1.2 L/min. Ultrafine and fine particles in the 0.01-0.193 μm diameter range are measured with the electrical mobility spectrometer while fine and coarse particles in the 0.253-35 μm diameter range are measured with the optical particle counter. In order to conduct accurate aerosol measurements while the





vehicle is in motion, an isokinetic stainless-steel sampling probe (model 1.152, Grimm Aerosol Technik) suitable for air velocities up to 25 m/s is set up on the roof of the car at a height of 50 cm above the car's roof (2.37 m above ground) in the forwards direction. This setup prevents contamination by the car's own exhaust situated at about 30 cm above ground.

The real-time geolocation, altitude and driving speed data are recorded using a Garmin GPS set up on the roof of the vehicle. Ambient temperature, pressure and relative humidity are monitored at 1-second time resolution with a meteorological probe (model Testo) installed on the roof of the vehicle. Additionally, an action camera is sometimes set up on the roof to record pictures of the sky, useful for posterior data analysis. A router with an external antenna is used to connect the mobile system to the Internet using a SIM card with 3G connexion. Data acquired during the mobile measurements is further transferred to

laboratory-based server for routine processing. An online platform to visualize measurements in real-time is under development.

## 3 Methodology

This section is divided into two parts. Section 3.1 details the pre-processing and data quality assurance procedures for lidar and sun photometer measurements. Comparisons with instruments from reference networks such as EARLINET (European

Aerosol Research Lidar Network) and AERONET (Aerosol Robotic Network) are included in this section. The principle of inversion algorithms used to derive aerosol microphysical and optical properties such as volume size distribution, extinction coefficient, effective extinction-to-backscatter ratio and mass concentration are described in Sect. 3.2.

### 3.1 Data processing and instruments intercomparison

In this section we discuss the corrections applied to lidar attenuated backscattered signals as well as procedures to ensure

data quality of mobile lidar and sun photometer measurements. Comparisons with other similar instruments in reference lidar and sun photometer networks are conducted in order to show the reliability of our measurements.

### 3.1.1 Lidar data pre-processing and quality assurance aspects

In order to obtain the total attenuated backscatter from lidar signals several instrumental corrections are applied: non-linearity of the detector, afterpulse correction, background sky radiance, overlap and range correction. These have been

previously described in (Mortier et al., 2013). However, improvements at instrumental level have been carried on, reducing the blind zone (caused by the afterpulse phenomenon) to $z_{min}$=180 m for vertical sampling. The afterpulse signal is measured at the beginning of each mobile measurement and subtracted from the total received backscatter signal. Micropulse lidars are unique, compact systems that can be easily deployed in remote locations or integrated on mobile platforms. The system's design causes the full overlap to be reached at ranges between 4-6 km (Berkoff et al., 2003; Campbell et al., 2002). Thus, in

order to exploit quantitatively the lidar signals, the overlap factor correction must be very properly assessed and applied.





In our study, the overlap factor has been determined using three methods: (i) the slope method (Kunz and de Leeuw, 1993) using horizontal profiles when the lower atmospheric layers can be considered homogeneous, (ii) the slope method using vertical profiles in fixed location under fair weather conditions with low aerosol loading and (iii) taking the ratio of CE370 micro-lidar signals to another calibrated lidar, LILAS (Bovchaliuk et al., 2016; Veselovskii et al., 2016), integrated in

EARLINET since 2015. Figure 2 shows the average overlap factor assessed using the three methods and its standard deviation as a function of range. The different sources of uncertainty of the overlap functions determined with each method are supposed to be independent so that the total uncertainty is computed by taking the square root of the quadratic sum of each overlap function uncertainty. The overlap correction factor is smaller than 0.1 below 0.3 km, and it reaches unity around 6.8 km. The coefficient of variation of the overlap correction is less than 10% above 2 km. Below 2 km it increases

by as much as 25% reaching the maximum at 0.18 km (minimum altitude considered for exploitable physical signal). Therefore, an uncertainty on the overlap function of 10% above 2 km increasing to the surface to about 25% at 0.18 km will be considered throughout this study. The overlap correction has been checked using stationary observations in the period 2016-2017 after different mobile campaigns. Considering measurements over a long time period allows checking the overlap correction's stability. By using different methods and different lidar measurements over time, one includes both the

systematic and random errors that could impact the overlap factor correction. This explains the high variability and means that all types of possible errors are taken into account. The mean overlap factor assessed using the three methods has been used for processing the lidar signals in this work. The uncertainty on the overlap factor correction has been propagated onto the derived aerosol properties.

After all instrumental corrections are applied, the background, overlap and range-corrected lidar signals are checked

according to Rayleigh-fit criteria used in EARLINET lidar quality assurance procedures (Freudenthaler et al., 2018). The relative deviations from the calculated Rayleigh signal fall below 5% between 5 and 13 km. Between 13 and 16 km the relative deviations reach 10%, which is the maximum acceptable limit. This means that vertical sounding up to 16 km can be reached with the CE370 micro-lidar at nighttime and under low aerosol loadings (AOD of 0.06 at 532 nm).

It should be noted that the same model of micro-lidar (CE370) has been used for routine continuous aerosol monitoring over

Lille, France and M'Bour, Senegal since 2006 and aerosol studies using micro-pulse lidar data are presented in several works (Léon et al., 2009; Mortier, 2013; Mortier et al., 2013, 2016).

### 3.1.2 Comparison with reference lidar in EARLINET

The performance of CIMEL CE370 micro-pulse lidar has been assessed by comparison with a multi-wavelength Raman lidar, IPRAL (Bravo-Aranda et al., 2016) operating at SIRTA (Site Instrumental de Recherche par Télédétection

Atmosphérique; Haeffelin et al., 2005), Palaiseau, France (48.7° N, 2.2° E; 156 m asl). IPRAL lidar system is part of EARLINET and undergoes the network's quality assurance procedures. The most interesting feature of IPRAL for this study is its newly integrated near-field telescope, which gives access to backscattering in the low altitudes of the atmosphere down to 300 m. The combined signals from the two telescopes (near-field and far-field) are used in this work and compared

against the signals of the mobile CE370 micro-lidar. The signals from IPRAL are adjusted to a common vertical resolution of 15 m and no vertical smoothing is applied to any lidar profiles. Night-time measurements on 28 August 2017 at Palaiseau, France are considered for comparison and range corrected signals (RCS) are averaged over 30 minutes between 23:15 and 23:45 UTC and normalized over a vertical range between 5.6 km and 7.6 km agl, where the aerosol content is considered

negligible. The comparison of normalized RCS at 532 nm from IPRAL and CE370 lidar, along with the molecular profile computed from radiosounding data at Trappes (48.76° N, 2.00° E; 168 m asl) at 00:00 UTC, 29 August 2017, are shown in Figure 3. The standard deviation of CE370 lidar RCS represented in Figure 3 as light green shaded area is the uncertainty on the overlap function.

A multi-layer situation was observed at Palaiseau at this time interval, which was an ideal case study for comparing the

performance of the two lidar systems. The profiles show a first aerosol layer from the ground up to 1 km agl along with a well-separated aerosol layer up to 4.5 km agl in the lower troposphere. In the upper troposphere and lower stratosphere cirrus clouds between 10.8 km and 12.2 km agl and an aerosol layer between 17 km and 20 km agl are observed. The air masses back trajectory analysis (not reported) shows the transport of desert dust in the 2-4.5 km range and most probably transported biomass burning particles at higher altitudes originating from intense forest fires in Canada (Khaykin et al., 2018).

The two profiles are remarkably similar showing very good agreement between the two systems despite the use of the overlap correction for the micropulse lidar data. The largest differences are observed in the overlap correction range of the CE370 lidar but the expected amplitudes of the signals fall within the overlap uncertainty, which means that our overlap correction is reasonable down to 0.6 km. The highest fractional differences between IPRAL and CE370 values of RCS are under 15% above 2.5 km and reach 70% at 0.18 km altitude (minimum altitude considered for CE370 lidar). The lowest

fractional differences are <5% above 0.84 km and up to 50% at 0.18 km. A good agreement between the two systems is observed with discrepancies that are within the CE370 lidar RCS uncertainty. For higher altitudes in the atmosphere, although the CE370 lidar's signal-to-noise ratio above 12 km is significantly lower compared to IPRAL system, the micropulse lidar is able to detect the aerosol layer in the UTLS (Upper Troposphere Lower Stratosphere) due to the strong backscattering signal of the layers. Applying noise filtering significantly improves the SNR and implies that backscatter

signals from far range could be exploitable in such rare situations.

This lidar comparison example is part of an intercomparison campaign that involved both research and micropulse lidars in ACTRIS-FR and METEO-FRANCE networks. This case study shows one side application of the mobile system within checking the uniformity of lidar measurements at different sites and the validation of micro-lidar measurements.

### 3.1.3    Sun photometer pre-processing and data quality

PLASMA sun photometer data is included in the AERONET database (instrument #650) and the instrument is calibrated by Service National d'Observation PHOTONS, French branch of AERONET, using Langley method at Izaña Observatory (28.3° N, 16.5° W; 2400 m asl) following the AERONET calibration protocol for the AERONET reference master instrument. PLASMA sun photometer is intercalibrated regularly against a CIMEL CE318 master sun photometer from



PHOTONS network at the Carpentras site (44.1° N, 5.1° E; 100 m asl). This allows checking the stability of the instrument as the characteristics may change over time. Figure 4 presents the comparison of spectral AOD from PLASMA and CIMEL sun photometers coincident measurements at Carpentras site, on 12 June 2017. The comparison with a master instrument shows excellent agreement between the two instruments with RMS AOD differences better than 0.005 for all channels.

Regarding data processing for mobile measurements, the data is filtered over time steps of 10 to 30 seconds with a criterion of 2% maximum difference for the measurements in the selected time interval. This filter rejects the artifacts from measurements through clouds or when passing by obstacles (trees, buildings, bridges, tunnels, etc.). Filtered measurements are then submitted manually to AERONET processing system to derive spectral AOD. However, this method takes some time, up to 1 day, to get the calibrated AOD. Therefore, the internal PHOTONS AOD computation processing has been

rather considered for campaigns to produce real time AOD. The desktop-based acquisition software shows the AOD not calibrated in real time, useful for checking PLASMA measurements along the road. This monitoring will be improved in order to produce AOD level 1.5 data (cloud screened) in the future STrAP (Système de Traitement des AOD de Plasma) web-based processing system. This will allow the visualization of real-time calibrated PLASMA AOD.

### 3.1.4    Aerosol spectrometer

Optical particle counters (OPC) similar to mini-WRAS aerosol spectrometer have been used for mobile applications such as aircraft, car and underground station measurements (Bush et al., 2015; Cheng and Lin, 2010; Weber et al., 2012). Grimm and Eatough (2009) have shown that the $PM_x$ mass concentrations obtained from the conversion of size distribution to mass distribution are in good agreement with FDMS measurements, equivalent to gravimetric measurements. Comparison of the GRIMM mini-WRAS particle counter with a conventional TEOM FDMS monitor from an air quality station situated nearby,

around 5 km distance from our laboratory, shows very good agreement between the two instruments, despite the different methods and different locations. This demonstrates the reliability of the derived $PM_x$ mass concentrations presented further in our study.

### 3.2 Retrieval of aerosol properties

Physical aerosol parameters measured include aerosol and cloud backscattered signals (lidar), solar spectral irradiance (sun

photometer) and particle concentration (aerosol spectrometer). The inherent aerosol optical properties (extinction coefficient) and the microphysical properties (volume size distribution) can be retrieved from coincident measurements using direct or inverse calculations. In this section we present the methodology employed to obtain lidar-sun photometer-derived extinction coefficient, effective extinction-to-backscatter ratio and mass concentration as well as sun photometer-derived volume size distribution from on-road mobile measurements.





### 3.2.1 Total column volume size distribution

We use the recently developed GRASP (Generalized Retrieval of Aerosol and Surface Properties) (Dubovik et al., 2014) algorithm and software (more information at http://www.grasp-open.com/) to derive total column aerosol volume size distribution from spectral direct sun photometer measurements. This particular application is called GRASP-AOD and has been described in detail by Torres et al. (2017). The algorithm is based on multi-term least square method; the retrievals start from a priori constraints on actual values and are performed until the residuals are minimized. Aerosols are modelled as a mixture of spherical and non-spherical fractions and the inversion requires an assumption on the refractive index, real part and imaginary part, as well as on the sphere fraction. Six parameters describing the bimodal lognormal size distribution are retrieved, median radius, volume concentration and standard deviation for both fine and coarse mode, as well as secondary aerosol properties such as effective radius, total concentration and fine mode AOD at 500 nm. In this work we show the application of GRASP-AOD inversion to the on-road PLASMA sun photometer measurements. The uniqueness of the algorithm lies in the fact that it does not need coincident sky radiance measurements and it can be used for applications such as mobile sun photometer measurements to determine the spatial variability of total column aerosol size distribution, under certain assumptions and known limitations.

### 3.2.2 Aerosol extinction coefficient profiles and effective extinction-to-backscatter ratio

The lidar backscattering signals contain information on atmospheric scattering and extinction processes. After all corrections detailed in Sect. 3.1.1 are applied on lidar signals, the lidar equation can be written as

$$S(z) = \left(\beta_{aer}(z) + \beta_{mol}(z)\right) exp\left\{-2 \int_{z_{min}}^{z} \left(\sigma_{aer}(z') + \sigma_{mol}(z')\right)dz'\right\}, \tag{1}$$

where $S(z)$ is the attenuated backscatter, that is, the lidar background, range, and overlap corrected, calibrated, energy normalized lidar signal. $\beta(z)$ and $\sigma(z)$ are the range dependant volume backscattering and extinction coefficients and the subscripts "aer" and "mol" refer to the contribution of aerosols and molecules, respectively. The lidar equation (Eq. 1) is an undetermined equation, with two unknown variables ($\beta_{aer}$ and $\sigma_{aer}$), so a relationship between the two variables is needed. The aerosol extinction-to-backscatter ratio or lidar ratio (LR) is introduced,

$$LR_{aer}(z) = \frac{\sigma_{aer}(z)}{\beta_{aer}(z)} \tag{2}$$

function which depends on the size distribution, wavelength, shape and composition of aerosols

$$LR_{aer} = \frac{4\pi}{\varpi_0 P(\pi)} \tag{3}$$

where $\varpi_0$ is the single scattering albedo and $P(\pi)$ is the phase function at scattering angle 180°.

The molecular extinction-to-backscatter ratio is constant, $LR_{mol}=8\pi/3$ sr, and $LR_{aer}$ must be assumed vertically constant in order to simplify the equation. The methodology to invert lidar signals is similar to that described by Leon et al. (2009) and is based on the Klett and Fernald (1981, 1984) solution to the inverse problem.





$$\beta_{aer}(z) + \beta_{mol}(z) = \frac{S(z)exp\left\{-2\int_{z_{ref}}^{z}[LR_{aer}(z)-LR_{mol}]\beta_{mol}(z)dz\right\}}{\frac{S(z_{ref})}{\beta_{aer}(z_{ref})+\beta_{mol}(z_{ref})}-2\int_{z_{ref}}^{z}LR_{aer}(z')S(z')exp\left\{-2\int_{z_{ref}}^{z'}[LR_{aer}(z'')-LR_{mol}]\beta_{mol}(z'')dz''\right\}dz'} \tag{4}$$

where LR is the extinction-to-backscatter ratio or lidar ratio; the subscripts "aer" and "mol" referring to aerosol and molecules lidar ratio. $z_{ref}$ is the reference altitude, where the signal is supposed to come only from molecular scattering.

Height-independent values of lidar ratio and extinction coefficient profiles are retrieved using an iterative inversion method
constrained by sun photometer AOD. A dichotomous approach is used on the LR values converging until the difference between lidar and sun photometer AOD at 532 nm is minimized. The same method applied on micro-lidar observations has been used in previous studies (Chazette, 2003; Chazette et al., 1995; He et al., 2006; Mortier et al., 2013) and proved to be reliable for deriving realistic LR values, that are consistent with calculated LR from AERONET measurements or with a Mie code. In this work mobile lidar profiles were inverted into extinction coefficient profiles using the constraint of coincident
AOD at 532 nm interpolated from PLASMA measurements.

### 3.2.3   Mass concentration profiles

From extinction coefficient profiles, mass concentration can be derived if assumptions on the atmospheric aerosols are imposed. The profiles of mass concentration are calculated using

$$M(z) = \frac{4}{3}\sigma_{aer}(z)\int_{r_{min}}^{r_{max}}\frac{\rho(r)\cdot r^3\cdot n_1(r)}{\int_{r_{min}}^{r_{max}}Q_{ext}(m,r,\lambda)\cdot n_1(r)\cdot r^2 dr}\, dr \tag{5}$$

where $r$ is the particle radius, $n_1(r)$ is the normalized volume size distribution, $Q_{ext}$ is the extinction efficiency and $\rho$ is the particle density. The full description of the methodology can be found in (Mortier et al., 2013), where it was applied for volcanic ash mass concentration calculations. The method requires knowledge of aerosol size distribution and refractive index and $Q_{ext}$ is computed using Mie theory. Climatological values for different aerosol types are considered for particle density ($\rho$). Practically, the sun photometer-derived size distribution is used, when possible, in order to construct an adequate
aerosol model for the atmospheric situation. In this work, the described method to derive aerosol mass concentration from lidar measurements has been applied to mobile observations. The parameters chosen for the calculations will be described for the considered case study in Sect. 4.1.

### 3.2.4   Discussion on uncertainties

The most important source of uncertainty on lidar profiles is the uncertainty on the overlap function, especially at lower
altitude levels where the incomplete overlap affects the measurements. Applying a wrong overlap correction can lead to an underestimation or overestimation of the attenuated backscatter signals and consequently of the derived variables such as extinction coefficient and mass concentration in the near-field range. Since most of aerosols are located near the surface and up to 1-2 km or up to 4-5 km in case of transported aerosols, the problem of incomplete overlap must be solved in order to quantitatively analyse aerosol properties within the first 5 km. An uncertainty of 10% above 2 km, increasing to 25% at





ground level, was assessed for the overlap correction factor used for our lidar data (Sect. 3.1.1). Additional uncertainties on lidar measurements are the statistical fluctuations of the measured signal, the detector dead-time, the fluctuations in laser energy and the afterpulse correction. According to Welton and Campbell (2002), the contribution of these corrections is either negligible or less then 5%.

The main sources of uncertainties in the retrieval of extinction coefficient come from the unknown lidar ratio vertical variation, the uncertainty on the lidar signal at the reference altitude, the uncertainty on the overlap function and the missing signal below $z_{min} \approx 180$ m. The molecular model can also induce a relative uncertainty of 5% according to Chazette et al. (1995). In our study we use the radiosounding data from the closest site when possible, trying to reduce the uncertainties related to the molecular profile. If the layers in the atmosphere are not well-mixed, the assumption of a constant lidar ratio

will lead to a bias in the retrieval of extinction coefficient profiles. Nonetheless, we constrain the retrievals with the integrated aerosol extinction (AOD), which improves the reliability of retrieved LR values, compared to the case when a LR is imposed. In the absence of direct measurements of LR vertical variation, column-averaged LR is the closest estimate that can be achieved. The overall error in the aerosol extinction coefficient and lidar ratio is not easy to be precisely computed, so in this study we estimate errors to be in the range of 15-25%, with maximum uncertainty at $z_{min} \approx 180$ m.

As we apply the same methodology as Mortier et al. (2013) to derive the mass concentration profiles for mobile measurements, the same uncertainties have been considered here. Considering an uncertainty of 15-25% on the extinction coefficient along with the uncertainty on the particle size distribution, refractive index and density, a total average uncertainty of 35-40% is expected on the mass concentration when considering independent errors.

The uncertainty on AOD from PLASMA sun photometer comes from the uncertainty in the calibration transfer from a

standard sun photometer. The uncertainty on PLASMA AOD in the visible and NIR is 2% compared to 1% for a standard CIMEL sun photometer and 3% compared to 2% in the UV.

Regarding the retrieved aerosol total column size distribution, the uncertainty on the fine mode volume median radius ($r_{Vf}$) and volume concentration ($C_{Vf}$) is between 5% for the fine-mode predominant cases and 10% for the coarse-mode predominant cases. The uncertainty on the retrieved coarse mode volume median radius ($r_{Vc}$) and volume concentration

($C_{Vc}$) is larger than 10% for the fine-mode predominant cases. For cases with coarse-mode predominance, the uncertainty is 10% for $r_{Vc}$ and around 20% for $C_{Vc}$, as shown by Torres et al. (2017). The characterisation of fine-mode is quite accurate even though reliable a priori information on refractive index is needed. The characterisation of the coarse mode is more difficult due to lack of information in this spectral range, but can be improved using moderate a priori information on coarse-mode parameters (for example, from near almucantar inversions).

**4 Results**

Two case studies are presented in this section, aiming to give examples of the products that the algorithms can provide and also to illustrate the applications of an instrumented mobile system. Section 4.1 focuses on remote sensing measurements





performed in the North of France on 26 August 2016, along the transect Lille-Dunkerque. Comparisons with satellite AOD data from Moderate Resolution Imaging Spectroradiometer (MODIS) Aqua daily product MYD04 and with simulations from Dust Regional Atmospheric Model (BSC-DREAM8b) are performed and presented. Section 4.2 presents in situ measurements performed along the route Lille-Paris on 28 August 2017. Comparison between on-road particle counter-

derived $PM_{10}$ and ATMO Hauts-de-France and AIRPARIF $PM_{10}$ measurements and modeled $PM_{10}$ concentrations are conducted and included in this section.

### 4.1 Remote sensing mobile measurements in northern France

### 4.1.1 Observation strategy and meteorological conditions

The MAMS mobile exploratory platform was deployed in northern France in the spring and summer periods of 2015-2017,

periods marked by higher occurrence of pollution events (Unga, 2017). During the same periods, long-range transport of aerosols over northern France region is quite frequently observed by our continuous measurements. The work of Mortier (2013) illustrates variability of aerosol events over Lille during 2006-2012 period.

We present here mobile measurements performed on 26 August 2016 on the route between Lille (50.61 °N, 3.14° E) and Dunkerque (51.03° N, 2.37° E), situated 80 km northwest from Lille. Mobile measurements were performed between 12:00

UTC and 13:07 UTC (local time = UTC +02:00) along Lille-Dunkerque route and between 14:20 and 15:52 UTC along Dunkerque-Lille route. The mobile measurements were triggered based on chemical transport model forecasts provided by the ESMERALDA platform (http://www.esmeralda-web.fr/) and PREV'air system (http://www2.prevair.org/) and based on BSC dust forecasts (http://www.bsc.es/ess/bsc-dust-daily-forecast). Daily maximum $PM_{10}$ concentrations at ground level exceeding 50 μg/m$^3$ were expected for Lille and lower concentrations around 30 μg/m$^3$ for Dunkerque. $PM_{10}$ concentrations

at ground level measured by air quality stations in Lille exceeded 100 μg/m$^3$ at 07:00 local time, which lead us to investigate this pollution event. The prediction maps showed that the pollution plume covered the Netherlands, Belgium and North of France regions, with an increase in particulate matter and $NO_2$ concentrations, which are indicators of anthropogenic pollution. Total $NO_2$ from OMI satellite measurements (http://www.temis.nl/airpollution/no2.html) indeed showed increase of concentrations over northern France, Belgium and Netherlands. The BSC DREAM8b model indicated transport of dust

over the coastal region, penetrating inland up to Lille and dust layers between 2 km and 5 km, of low concentration, were predicted for Lille. Regarding the meteorological conditions, anticyclonic conditions maintained a dry and sunny weather over North of France, with near surface temperatures between 26° C and 28° C at Lille and around 21° C at Dunkerque and with low wind speeds in the range of 11 to 13 km/h from N-NE direction.

The main goal of our measurements was to reveal the spatial variability of atmospheric structures and the evolution of the

aerosol optical properties (aerosol extinction coefficient) along the route, away from Lille agglomeration. The existence of a major motorway axis between Lille and Dunkerque makes it possible to sample this region easily and quickly.




### 4.1.2 Spatial variability of AOD and comparison with MODIS data

A strong variability was observed between the two end points, Lille and Dunkerque. High AOD values in the range 0.6 - 0.8 at 440 nm were recorded at Lille and surroundings between 11:00 UTC and 12:00 UTC and decreased rapidly to 0.37 when arriving at Dunkerque around 13:07 UTC. The Ångström exponent, between 1.4 and 1.6 at Lille, decreased to 1.2 along the

route to Dunkerque. The values recorded at Lille are characteristic for fine particles, typical for urban sites, while the decrease of the Ångström exponent along the road indicates the presence of larger particles. During one hour of stationary measurements starting from 13:15 UTC, the AOD levels remained stable around 0.4. On the way back to Lille, AOD and Ångström exponent values increased to 0.75 and 1.6, respectively. AOD at 440 nm and Ångström exponent from PLASMA measurements were compared with data from closest AERONET sites (Lille and Dunkerque) and a mean absolute difference

of 0.04 was found, proving the reliability of PLASMA measurements. One must take into account that for this comparison the closest measurements (in space and time) were considered and that they are not taken at the exact same location as AERONET sites.

As an example of the applications of the mobile system, MODIS AOD data has been evaluated by comparison with ground mobile sun photometer measurements. We used the MODIS deep blue (DB) product at 10 km over land (Levy et al., 2013)

for the comparisons. MODIS AOD product with higher resolution (3 km) would have been preferred to better address pollution gradients, but not enough pixels coincident with our mobile transect were available for this day. Figure 5 presents satellite AOD retrievals at 550 nm from MODIS Aqua daily product MYD04 along with AOD at 550 nm from mobile PLASMA sun photometer measurements. PLASMA data were spectrally interpolated to 550 nm using the standard Ångström exponent method. The overpass of Aqua satellite over the region is at 12:05 UTC, so we chose the route Lille-

Dunkerque as representative for our analysis, as mobile measurements started around 12:00 UTC. The mean sun photometer AOD was obtained by averaging AOD data that fell in each MODIS pixels. Five sets of PLASMA-MODIS AOD were considered for comparison within the sampling box.

Having a larger spatial coverage, MODIS data shows higher AOD values along the coast and over northern France and Belgium-Netherlands regions, in the range 0.3 and 0.8, which is consistent with model predictions of the pollution event and

of the dust transport. As we can see, PLASMA and MODIS data are in good agreement. MODIS was highly correlated with ground mobile sun photometer with $R^2$ of 0.76, slope of 1.13, intercept of 0.11 and RMSE of 0.17. These results are consistent with the findings of Wang et al. (2017), $R^2$ of 0.76, slope of 0.9, intercept of 0.11 and RMSE of 0.17 for Aqua-MODIS AOD at 3 km (MYD04_3K). The results of comparison are very good considering the uncertainty due to the atmospheric variability imposed by atmospheric motion and different times of MODIS and PLASMA measurements. The

lack of AOD data in the same regions for both instruments is also consistent and is due to the presence of scattered clouds at 4 km altitude, showing that both cloud-screening algorithms are successful.


### 4.1.3 Analysis of LiDAR vertical observations

Figure 6 shows the lidar range-corrected signals (RCS) recorded along the route Lille-Dunkerque (Figure 6a) and Dunkerque-Lille (Figure 6b), respectively. High aerosol backscatter is observed up to 1 km altitude at Lille, explained by an on-going particle pollution event, decreasing towards the coastal region (Dunkerque). According to the chemistry-transport model predictions, the pollution event impacted Lille city and surroundings within 30 km distance with predicted $PM_{10}$ levels exceeding 50 μg/m$^3$, while a gradient in $PM_{10}$ concentrations was expected when moving westward, towards the coast. This is consistent with our lidar observations, showing a decrease in the aerosol backscatter in the first aerosol layer (from surface up to 1 km) approximately 30 km away from Lille and even a stronger decrease when approaching Dunkerque. This gradient was observed with our mobile measurements during the whole day, for both transects. The PBL height decreased from 1 km at Lille to 0.6 km at Dunkerque, showing the contrast between continental and coastal sites. Moreover, outside Lille region, the presence of several aerosol layers up to 5 km is revealed. These layers are hardly observed over Lille due to the strong backscatter in the lower altitudes, which strongly attenuates the laser beam. Dust aerosol layers between 2 km and 5 km were observed by lidar measurements as confirmed by Dust Regional Atmospheric Model (DREAM, http://www.bsc.es/ projects/earthscience/DREAM). Lower Ångström exponent along the road and at Dunkerque indicate an increase of the aerosol coarse-mode fraction and the analysis of backward trajectories performed with HYSPLIT (Hybrid Single Particle Lagrangian Integrated Trajectory) model confirm dust transport over northern France.

### 4.1.4 Sun photometer-derived volume size distribution

Total column aerosol volume size distributions retrieved with GRASP-AOD are presented in Figure 7. At the left part of Fig. 7 spectral AOD from PLASMA and AERONET sun photometer measurements at Lille (15:31 UTC) and Dunkerque (13:54 UTC) are shown. The GRASP-AOD inversions and AERONET standard inversions are illustrated in the middle and right part of Fig. 7 for Dunkerque and Lille, respectively. Note that PLASMA measurements are performed at slightly different location and time, compared to AERONET data. The fine mode volume median radius ($r_{Vf}$) is in the range of 0.18 μm to 0.16 μm for Lille and Dunkerque, respectively, while $r_{Vf}$ from AERONET standard inversion is 0.18 μm and 0.20 μm, respectively. For coarse mode volume median radius ($r_{Vc}$) values of 2.25 μm and 2.17 μm are obtained, while AERONET gives 2.48 μm and 2.30 μm, for Lille and Dunkerque, respectively. The concentration values of the fine mode obtained for Dunkerque are: $C_{Vf}$ = 0.04 μm$^3$ μm$^{-2}$ (GRASP-AOD) and $C_{Vf}$ = 0.04 μm$^3$ μm$^{-2}$ (AERONET) and of the coarse mode are: $C_{Vc}$ = 0.07 μm$^3$ μm$^{-2}$ (GRASP-AOD) and $C_{Vc}$ = 0.06 μm$^3$ μm$^{-2}$ (AERONET). For Lille, $C_{Vf}$ = 0.08 μm$^3$ μm$^{-2}$ is retrieved with GRASP-AOD compared to $C_{Vf}$ = 0.12 μm$^3$ μm$^{-2}$ from AERONET inversion while for coarse mode, $C_{Vc}$ = 0.07 μm$^3$ μm$^{-2}$ from GRASP-AOD and $C_{Vc}$ = 0.06 μm$^3$ μm$^{-2}$ from AERONET. We should note that the AERONET inversion for Lille does not pass the Level 2 criteria of Version 3 that requires at least 3 measurements to be available for scattering angles equal or higher than 80°. This can induce some bias in the retrievals of size distribution.



An a priori refractive index of 1.46-0.003i was used for the inversions, choice made taking into account the closest AERONET standard inversions and the predominant aerosol types within the atmospheric column. A good consistency between the volume size distribution from AERONET standard inversion and GRASP-AOD inversions is generally observed taking into account the uncertainties discussed in Sect. 3.2.4. Figure 8 illustrates the volume size distribution from

GRASP-AOD inversions along Lille-Dunkerque transect (12:18 UTC – 13:07 UTC). Few sun photometer measurements were possible at the departure time (12:00 UTC) due to the presence of scattered clouds and are not shown, but higher amplitude of fine mode fraction is observed at Lille and close to Lille, decreasing towards Dunkerque. The fine-mode predominant size distributions show that the fine particles pollution episode is localized over Lille region while the coarse mode contribution comes partly from dust transport. This is a unique illustration of the spatial variability of column size

distribution as retrieved with GRASP-AOD from mobile sun photometer measurements.

**4.1.5 Aerosol extinction coefficient profiles and extinction-to-backscatter ratio**

Figure 9 presents the spatio-temporal variability of extinction coefficient profiles along the route Lille-Dunkerque retrieved by applying Klett inversion algorithm to lidar data constrained by coincident AOD. The upper range for the reference signal is taken between 5 km and 5.5 km where the aerosol contribution is considered negligible. This has been checked against the

molecular profile calculated using radiosounding data on 26 August 2016, 12:00 UTC from Trappes (48° 45' N, 2° E). In this study, the extinction coefficient profile is extrapolated from $z_{min} \approx 180$ m to the ground level.

The contrast in extinction coefficient between surroundings of Lille and towards the coastal site is striking. Mean and maximum extinction of 0.13 km$^{-1}$ and 0.33 km$^{-1}$, respectively, for the first layer up to 0.7 km, are found close to Lille, while values of 0.05 km$^{-1}$ and 0.14 km$^{-1}$ for mean and maximum extinction are found along the second half of the road. A second

layer is revealed between 1.7 km and 2.2 km, with mean and maximum extinction of 0.17 km$^{-1}$ and 0.54 km$^{-1}$, respectively. The fine layers between 2.5 km and 5 km show lower contribution to the total aerosol loading with mean and maximum extinction of 0.04 km$^{-1}$ and 0.10 km$^{-1}$, respectively.

The column-integrated lidar ratio (LR) ranges from 35 (±7) sr to 60 (±14) sr and the average value is close to 43 (±14) sr. Higher LR are found close to Lille, between 51 (±10) sr and 60 (±14) sr. The LR have been also calculated with a Mie code

using the GRASP-AOD derived size distribution and refractive index associated for inversions and the LR values, in the range 40-49 sr, are in good agreement with the lidar-derived LR. The LR retrieved from lidar data have been also compared to values derived from standard CIMEL sun photometer measurements (Léon et al., 2009) and were found to be on average 20% lower, which was observed also in our case when comparing our results with LR calculated from AERONET standard inversions. However, the values of LR from standard sun photometer lie within the uncertainty on LR caused by the

uncertainty on overlap correction.

The fine dust layers above 2.5 km have a small contribution to the column effective LR, so the values obtained can be attributed to the aerosol layers below. The values found are characteristic to small, absorbing anthropogenic particles with



LR values typically between 50 and 80 sr (Ackermann, 1998). The values of LR are in good agreement with other studies on coastal sites: 32-63 sr in Sagres, Portugal (Ansmann et al., 2001) and 33 (±14) sr to 65 (±15) sr at Dunkerque, France, during sea breeze events (Boyouk et al., 2011). In our case, the decrease in LR when approaching Dunkerque also suggests a change from urban aerosol type to a more marine type. A shallow layer at 400 m causes a strong backscatter, suggesting that this is

the marine boundary layer. The height-resolved lidar profiles indicate that the layers of marine and continental particles are well delimited, probably due to a stable stratification of the lowermost layers prohibiting the mixing between different aerosol types. We suppose that particles in the second layer, between 1.7 and 2.2 km, are of different nature than the particles in the layers above, and that they are more absorbing and hygroscopic. This hypothesis is based on stationary observations at Dunkerque that showed possible water uptake by particles in this layer, leading to an increase of the particle size explained

by a decrease in Ångström exponent and a rapid increase in the backscatter coefficient. If the relative humidity at this level is close to saturation this leads to a large increase in the aerosol extinction coefficient. However it is not possible to conclude on this effect without the actual profile of relative humidity. These are only hypotheses that could be clarified with more information on particle size and shape, using a 2-λ lidar and polarization channel (CIMEL, model CE376), planned to be integrated in the mobile system's payload.

**4.1.6 Comparison with BSC-DREAM8b at fixed location**

In our study, we evaluate the BSC-DREAM model simulations over Lille by intercomparison with ground-based lidar-sun photometer measurements. Figure 10 shows the comparison between the aerosol extinction coefficient and mass concentration profiles at 532 nm derived from lidar-sun photometer mobile measurements near Lille, at 15:30 UTC, and the dust extinction coefficient and concentration profiles at 550 nm from BSC DREAM8b simulations for Lille, 16:00 UTC. The

uncertainty due to the overlap correction is represented with light shaded area. The extinction AOD at 532 nm for the mean profile in Figure 10a is 0.53 and the lidar ratio is 66 (± 14) sr, close to the column integrated lidar ratio (59 sr) derived from AERONET standard inversion at Lille, 15:30 UTC.

We first compared the aerosol extinction coefficient at the lowest lidar detectable range (180 m) with the extinction coefficient at ground level computed at 532 nm using Mie theory. Aerosol scattering and absorption coefficients can be

calculated if the size distribution and refractive index are known and assuming spherical particles. For our Mie calculations we used the size distribution measured at Lille, 15:30 UTC by an aerosol spectrometer and the refractive index was computed using an indirect method. For the last nephelometer and aethalometer measurements at Lille, 08:00 UTC, aerosol absorption and scattering coefficients were simulated for different refractive indices until the difference between measured and computed scattering and absorption coefficients was minimized. A value of 1.58-0.01i was found for the complex

refractive index. This value is within the range of retrieved refractive indices of 1.56-0.01i to 1.58-0.01i found by Levin et al. (2009) for a mixture of organics, soil, sulphates, nitrates and carbon, which are characteristic components in urban environments (Niemi et al., 2006). After finding the refractive index, we apply Mie theory to compute aerosol extinction



coefficient at 532 nm. We can see in Fig. 10a an excellent agreement between the calculated extinction coefficient (0.10 km$^{-1}$) at ground level and the near surface (180 m AGL) lidar-derived extinction coefficient (0.10 ± 0.03 km$^{-1}$).

In order to compare our observations to the BSC's DREAM8b dust model simulations, the lidar extinction coefficient profile was reduced to the model's vertical resolution by applying sliding averages around model's height levels. The dust layer is

delimited between 2 and 5.8 km according to both observations and model. The mean lidar-derived extinction is 0.025 km$^{-1}$ (±0.015), while the model's mean extinction is 0.012 km$^{-1}$ (±0.006), resulting in a mean bias of -0.01 and a RMSE of 0.02. For the comparison, it should be kept in mind that the lidar-derived extinction coefficient uncertainties are of the order of 15-25%. The optical depth of the dust layer is 0.08 at 532 nm from lidar observations compared to 0.04 at 550 nm from model simulations. The differences between model and observations can be explained by the limited vertical model resolution

compared to the lidar resolution and by the spatial and temporal differences between the observed and modelled profiles considered for the comparison. The contribution of anthropogenic pollution below 2.5 km explains the differences between model and observations, since the model provides extinction coefficient only for dust particles. Mobile observations between Lille and Dunkerque show that the predictions of the dust model is correct, even for low AOD of the dust layers. A strong peak in extinction is observed at 870 m, with value as high as 0.69 km$^{-1}$ (±0.02). Radiosonde measurements at Trappes at

12:00 UTC (not shown here) show a PBL height of the order of 878 m, marked by a maximum of relative humidity of 73%, which agrees surprisingly well, despite different locations, with the strong increase in extinction coefficient at 870 m AGL observed from lidar measurements over Lille. Urban aerosols are subject to water uptake (Swietlicki et al., 2008), which leads to changes in aerosol optical properties that could explain the high extinction values at the top of the PBL.

The lidar-derived mass concentration profiles (Fig. 10b) were calculated from the retrieved extinction coefficient and by

assuming typical aerosol properties. A first aerosol model, constituted of a fine-mode dominant volume size distribution and a refractive index of 1.58-0.01i, was applied for calculations of mass concentrations up to 2.4 km altitude, while above 2.4 km, a size distribution with equal contributions of fine and coarse fractions and a complex refractive index of 1.5-0.005i were used. These choices were made based on hypotheses on the aerosol types within the atmospheric column: small, absorbing particles in the first layers up to 2.4 km and less absorbing, transported dust particles above. The aerosol density

values used for the calculations were 1.7 g cm$^{-3}$ and 2.6 g cm$^{-3}$, relevant for urban and desert dust particles, respectively. A mean lidar-derived mass concentration of 16 ± 10 µg/m$^3$ is obtained for the dust layer above 2.4 km, compared to a mean modelled dust concentration of 11 ± 6 µg/m$^3$, resulting in a mean bias of -5 and a RMSE of 9. These results show a pretty good agreement in regards of the uncertainties on both observations and model sides. The calculated mass concentration for the urban aerosols below 2.4 km reaches a maximum of 152 ± 5 µg/m$^3$ at 870 m and a value of 23 ± 6 µg/m$^3$ at near surface

(180 m AGL). The near-surface value is comparable to the closest PM$_{10}$ from ATMO Hauts-de-France (Lille Fives) air quality measurement, of 30 µg/m$^3$ at 15:45 UTC.





### 4.1.7 Mass concentration profiles

Figure 11 presents vertically-resolved mass concentration of aerosols retrieved from mobile measurements along Lille-Dunkerque route (same as in Sect. 4.1.5) using the optical-to-mass relationship defined in Sect. 3.2.3 and the microphysical properties defined in Sect. 4.1.6. We can notice that the concentration in the PBL is lower along the route and at Dunkerque

(12:36-13:08 UTC), mean and maximum of 28 µg/m$^3$ and 77 µg/m$^3$, respectively, than at Lille, mean and maximum of 77 µg/m$^3$ and 175 µg/m$^3$, respectively. Mass concentrations of the layer between 1.7 and 2.2 km reach a maximum of 287 µg/m$^3$ and a mean value of 90 µg/m$^3$. Higher concentrations in this layer can be explained by an increase in the extinction coefficient. Dust layers aloft 2.5 km show mass concentration of 26 µg/m$^3$ and 62 µg/m$^3$ for mean and maximum, respectively. Finally, we compared the lidar-derived mass concentration at the lowest detectable lidar range, located at 180

m, with PM$_{10}$ at ground level from two ATMO Hauts-de-France air quality stations, at Lille (Fives) and Dunkerque (Malo-les-Bains). The values from ATMO Hauts-de-France correspond to an averaged value over the last 15 minutes, while closest measurement in time was selected for mobile measurements. At Lille, a value of 58 µg/m$^3$ was measured at 12:30 UTC, while the lidar-derived mass concentration at 12:21 UTC, 30 km from Lille, was 77 µg/m$^3$. At Dunkerque, the station recorded a PM$_{10}$ value of 13 µg/m$^3$ at 13:15 UTC compared to 14 µg/m$^3$ from lidar-derived mass concentration calculations.

Taking into account all the hypotheses on the aerosol microphysical and optical properties and the uncertainty on the measurements, a maximum uncertainty of 40% has been previously defined. One can notice that in this case the relative difference between the lidar-derived mass concentration and the air quality measurements does not exceed 30%.

### 4.2 In situ mobile measurements and comparison with modelled PM$_{10}$

In this part we illustrate how the aerosol spectrometer mobile measurements reveal the spatial variability of the particle

concentration at the surface level. These measurements are complementary to the ones of lidar because they are carried out at ground level, in the lidar's "blind zone" (from surface to 180 m AGL). Additionally, it allows verifying the order of magnitude of the particle mass concentration derived from lidar-photometer coupling at its closest point to the surface. Moreover, these measurements can also be considered in a joint retrieval as an additional constraint to improve the extinction coefficient, mostly in the lower part of the profile (not done in this work). Mobile measurements of particle

number concentration have been performed along Lille-Palaiseau transect on 28 August 2017, from 08:00 UTC to 11:30 UTC. From these measurements, PM$_{10}$, PM$_{2.5}$ and PM$_1$ concentrations along the transect have been calculated. Predicted PM$_{10}$ maps given by the ESMERALDA (EtudeS MultiRegionALes De l'Atmosphere) platform (http://www.esmerald-web.fr/), which is based on the chemistry-transport CHIMERE model (Menut et al., 2013) have been considered in this study for comparison with PM$_{10}$ from mobile observations. The model provides hourly predictions of PM$_{10}$ (along with other

chemical components) on a 3-km resolution grid (regional domain) for each day. For our study, the model outputs assimilating the data from regional air quality stations were considered. In order to compare the PM$_{10}$ variability from model to that from mobile observations, we selected the modelled PM$_{10}$ corresponding to each hour spent on the road and the



associated section of the mobile transect. Figure 12a depicts the spatial variability of modelled $PM_{10}$ along with measured $PM_{10}$ on Lille-Palaiseau route. The ATMO Hauts-de-France and AIRPARIF air quality stations closest to the investigated route are also shown in Figure 12a. The model outputs and the mobile observations are consistent in describing the same gradient in $PM_{10}$ from Lille to Paris, showing concentrations higher than 65 µg/m³ around Lille and around 20 µg/m³ at

Palaiseau. Some differences can be noted, higher concentrations from mobile measurements than from model around Lille and localized peaks along the transect and when crossing Paris ring road. One must consider that our measurements have higher resolution (1 minute), while the model outputs correspond to the concentrations at the exact hour.  Spatially, we get localized measurements along roads compared to the 3-km resolution of the model. Furthermore, the mobile measurements are performed along highways, which can present higher variability of the concentrations, depending on the fluctuations of

the road traffic. The measured $PM_{10}$ when entering the highway at Lille reach 110 µg/m³ and decrease to 40 µg/m³ at 80 km away from Lille, and to 25 µg/m³, when approaching Île-de-France. Higher concentrations in the range of 43-72 µg/m³ are observed along the east side of the ring-road surrounding Paris. High levels of fine particles ($PM_1$) around Lille and the Paris ring road indicate the influence of heavy traffic. Table 1 reports the $PM_{10}$ levels recorded as 15-minutes average by air quality stations along the route and the corresponding 1-minute aerosol spectrometer measurements from mobile

observations. We checked the $PM_{10}$ values from air quality station, model and our measurements at the departure point, Lille, and the agreement is excellent: 62 µg/m³ measured by Lille Fives air quality station (average between 07:00-08:00 UTC), 61 ± 6 µg/m³ measured by the aerosol spectrometer on board the mobile platform (average between 07:30-08:00 UTC) and 61 µg/m³ from model (at 08:00 UTC). The mobile measurements were performed along the major highway (A1) connecting Lille to Paris, which explains the enhanced $PM_{10}$ values compared to the levels measured by regional air quality

monitoring stations. Taking into account all the variables, e.g. distance between the mobile and stationary measurements, type of station and temporal resolution, the $PM_{10}$ concentration levels from air quality stations and our measurements are in good agreement. Regarding the comparison with the modelled $PM_{10}$, leaving aside the different temporal and spatial resolution of the model versus observations, the results are in good agreement. As ESMERALDA models the background concentrations (due to the size of the mesh), the differences observed along highways are normal, especially as higher

concentrations along roads are dependent on unpredictable events like traffic jams. Our results show a possible application of the mobile platform for evaluating chemistry-transport models performances.

The aerosol extinction coefficient near surface (Fig. 12b) was obtained by applying scattering theory (Mie in our case) on the measured size distributions assuming a refractive index of 1.58-0.01i. Extinction as high as 0.50 km⁻¹ was obtained on the highway at Lille, which decreased to 0.07 km⁻¹ at Palaiseau. As mentioned previously, this information (ground level size

distribution-derived extinction coefficient) can be used to constrain the retrieval of extinction coefficient profiles at near surface level and will be implemented in a future version of our processing system.


## 5 Conclusions

This paper describes MAMS (Mobile Aerosol Monitoring System), a ground-based, lightweight, mobile exploratory platform dedicated to the measurements of atmospheric aerosols properties, designed for both stationary and mobile measurements during the vehicle's motion. Its uniqueness consists in combining remote sensing and in situ instruments for

investigating the vertical and spatial variability of aerosol properties. At this time, no other atmospheric mobile laboratory combines lidar, sun photometer and aerosol spectrometer. Measurements on-board include profiles of attenuated backscatter, extinction AOD, particle size distribution in the range of 0.01-30 µm and mass concentrations ($PM_1$, $PM_{2.5}$, $PM_{10}$) at surface level. Aerosol properties such as total column volume size distribution, extinction coefficient and mass concentration profiles as well as extinction coefficient at ground level are derived from the synergy of different measurements. Generally,

the participation of two operators is advised for performing mobile campaigns. The system being quite autonomous, only one driver is sometimes sufficient since follow-up of the on-going measurements can be achieved through remote access software. The advantage of the described mobile system is its great flexibility, being able to be deployed for on-road measurements with no or little preparation beforehand, compared to airborne campaigns that require more administrative permissions and that are more difficult to set up. In addition, the system is cost-effective compared to the organization of

extensive campaigns, which require more financial and human resources. Owing to the ease of its operation, the instrumented van can be deployed in case of sudden events necessitating fast reactivity, e.g. pollution and fire events, transport of dust, intrusion of volcanic ash etc. as well as for collecting vertical and surface data, important for modelling.

In this paper, the performance of the remote sensing instruments on-board the vehicle has been assessed by comparison with instruments in reference networks such as AERONET and EARLINET. Uncertainties has been also evaluated and discussed.

Over the period 2016-2017, more than 20 mobile campaigns in France and one collaborative campaign in North China Plain have been conducted. In this work we present two case studies meant to show the capabilities and applications of the mobile system. The first case study is focused on the mobile measurements between a continental (Lille) and a coastal (Dunkerque) site, when variations of AOD at 440 nm from 0.4 to 0.8, along with a high variability of aerosol structures vertically, over a distance of 80 km travelled in around 1 hour, have been recorded. Dust transport in the range 2-5 km altitude, added to a

significant local pollution event at surface level, illustrate an interesting case of spatial and vertical variability. This case exemplifies the use of the synergy of lidar-sun photometer measurements. The second case study, focused on in situ data and comparison with air quality model, showed a clear horizontal gradient in $PM_{10}$ concentration between Lille and Paris, which was consistent in both observations and model.

The utility of the mobile system for applications such as intercomparison with other lidars and sunphotometers in operational

networks, i.e. EARLINET and AERONET, has been shown. The validation of satellite-derived products, e.g. MODIS AOD, has been exemplified through comparison with mobile sun photometer measurements. Moreover, the use for validation of chemistry-transport models outputs has been illustrated through comparisons at fixed location (Lille) between profiles of



lidar-sun photometer derived parameters and dust model products. In addition, the assessment of mobile measurements of PM$_{10}$ versus modelled PM$_{10}$ at surface level showed interest for air quality model validation.

Near real-time visualization tools of all data sets are under development. On the one hand, data is planned to be displayed on a dedicated webpage and on the other hand a desktop-based analysis software (iAAMS – Automatic Aerosol Monitoring

Station) will collect all the measurements and display AOD, Angstrom Exponent, lidar RCS and extinction coefficient variability during field campaigns. Nevertheless, complex case studies remain a challenge and require post-analysis. The mobile system is versatile, providing flexibility in adding other instruments to the vehicle for specific campaigns. The integration of a second microlidar with two wavelengths and polarization channel is envisaged, which will help in the analysis of complex aerosol situations. Furthermore, sky-scanning possibility will be integrated as feature for the next

generation PLASMA sun photometer. Once these upgrades will be completed, the use of more advanced algorithms such as GRASP/GARRLiC (Lopatin et al., 2013) will give access to the vertical separation of aerosol fine and coarse modes and of absorption properties.

**Author contributions statement**

P.G., S.V., T.P., L.B. and R.L. conceived the concept of the mobile system. R.L. designed and implemented mechanical adjustments to the vehicle. P.G., I.E.P., T.P. and L.B. conceived the experiments. I.E.P conducted the experiments, analysed the data, prepared the figures and wrote the manuscript. T.P., L.B. and F.U. conducted the experiment(s). P.G. and S.V. supervised the work and contributed to the writing of the manuscript. C.Deroo and F.D. supported the data processing. A.M. developed the lidar-sun-photometer inversion code and I.E.P contributed to its evolution. B.T. provided expertise in GRASP-

AOD inversion code. C.Delegove developed the PLASMA acquisition software and helped solving technical issues. M.C. provided the aerosol spectrometer instrument. N.P.S. provided the air quality model data. C.P. provided the IPRAL lidar data. All authors reviewed the manuscript.

*Competing interests*. The authors declare that they have no conflict of interest.

*Acknowledgements*

We wish to thank ANRT France and CIMEL Electronique for supporting the research development on the mobile system and the CaPPA project (Chemical and Physical Properties of the Atmosphere) for the financial support. CaPPA project is funded by the French National Research Agency (ANR) through the PIA (Programme d'Investissement d'Avenir) under

contract ANR-11-LABX-0005-01 and by the Regional Council Hauts-de-France and the EU. We also thank ATMO Hauts-de-France and AIRPARIF for the PM$_{10}$ data, Service National d'Observation PHOTONS/AERONET from INSU/CNRS for the Cimel CE318 sun photometer data processing and calibration and we thank Sara Basart from BSC for providing the dust model data. The ACTRIS-FR research infrastructure is also acknowledged for financial support.



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



*Table 1: Comparison of measured PM₁₀ from air quality stations closest to the investigated route and the corresponding mobile measurements*

| Air quality (AQ) station | Station type | UTC AQ | $PM_{10}$ AQ $(\mu g/m^3)$ | UTC Mobile | $PM_{10}$ Mobile $(\mu g/m^3)$ | Distance (km) |
|---|---|---|---|---|---|---|
| Lille Fives (50.63° N, 3.09° E) | urban | 8:00 | 62 | 7:30-8:00 | 61±6 | 4 |
| Douai Theuriet (50.38° N, 3.07° E) | urban | 8:30 | 68 | 8:32 | 94 | 6 |
| Saint Laurent Blangy (50.31° N, 2.81° E) | suburban | 10:00 | 54 | 8:37 | 70 | 6 |
| Nogent sur Oise (49.28° N, 2.48° E) | urban | 10:15 | 27 | 10:18 | 33 | 16 |
| Creil (49.26° N, 2.47° E) | urban | 10:15 | 31 | 10:19 | 36 | 15 |
| Bobigny (48.904° N, 2.46 ° E) | background | 11:00 | 20 | 10:48 | 43 | 6 |
| Route nationale 2- Pantin (48.902° N, 2.39° E) | traffic | 11:00 | 40 | 10:48 | 43 | 1.2 |
| Boulevard Périphérique Est (48.84° N, 2.41° E) | traffic | 11:00 | 33 | 10:58 | 62 | 0.2 |

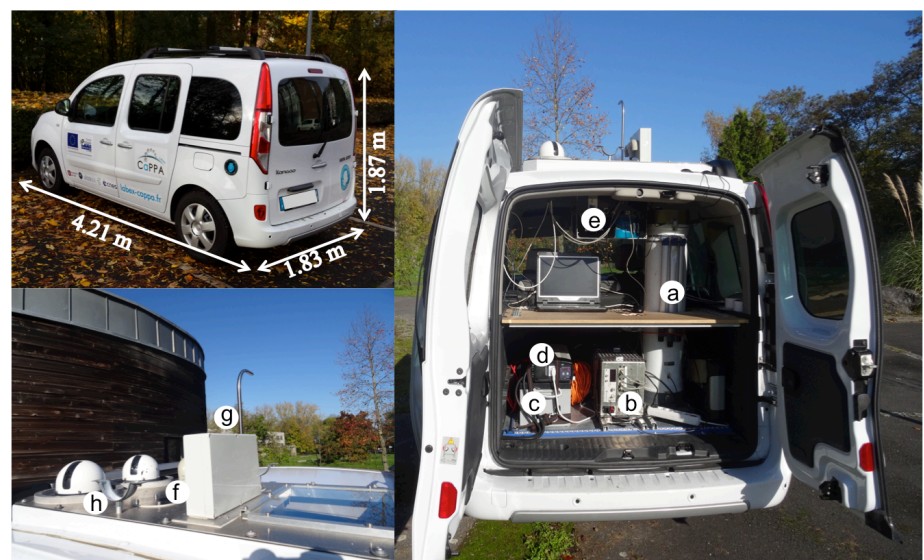

5 **Figure 1. Mobile system and equipment: (a) LiDAR transmitter-receiver optical head, (b) LiDAR control and acquisition unit, (c) battery, (d) sine-wave inverter charger, (e) aerosol spectrometer, (f) meteorological probe, (g) isokinetic sampling probe and (h) PLASMA sun photometers.**





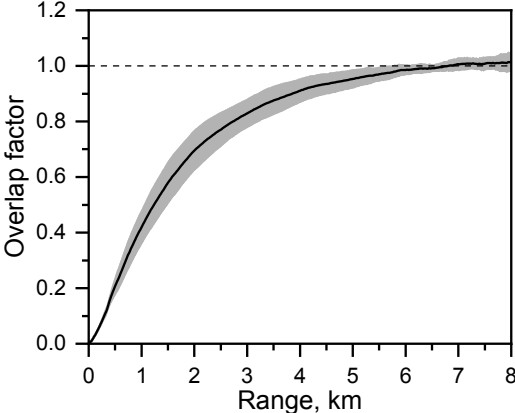

**Figure 2. Overlap factor of the CIMEL CE370 LiDAR on-board the vehicle versus distance. The combined standard deviation over the three methods used to assess the overlap function correction is represented by the light grey shaded area.**

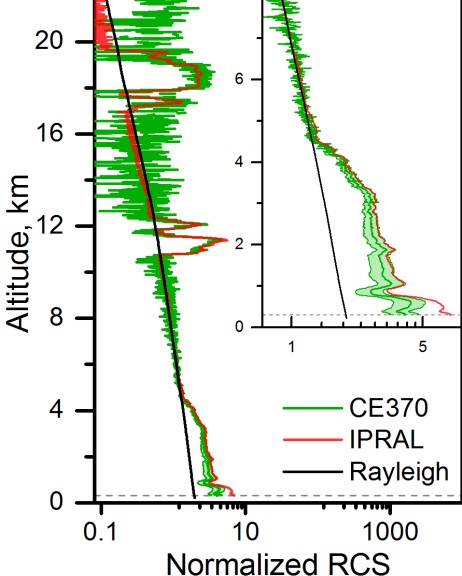

5    **Figure 3. Comparison of the normalized range-corrected signals profiles at 532 nm recorded by CIMEL CE370 lidar on-board the mobile platform (green) and IPRAL lidar (red) at Palaiseau, France, on 28 August 2017 (23:15-23:45 UTC). The profiles are displayed from 300 m above (complete overlap altitude of IPRAL system). The Rayleigh profile calculated from radiosonde measurements at Trappes on 29 August 2017, 00:00 UTC is represented by the black line.**

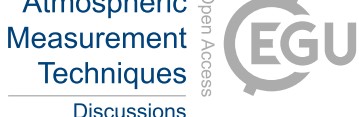



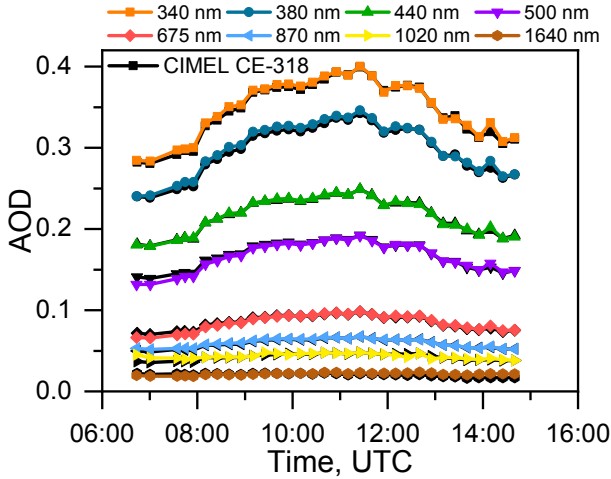

Figure 4. Comparison of the spectral extinction AOD from PLASMA (colored lines) and CIMEL CE-318 (black lines) sun photometers measurements on 12 June 2017 at Carpentras, France.

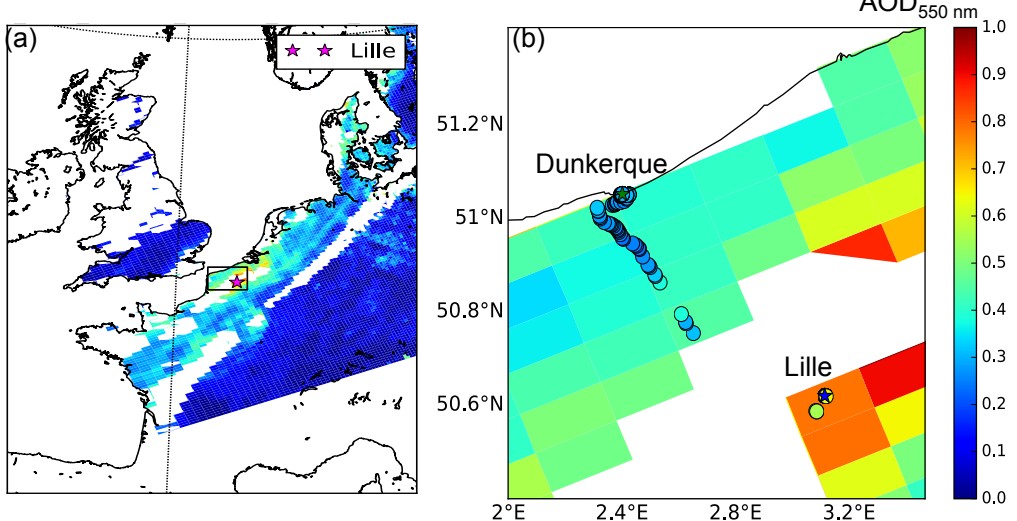

5   Figure 5. Spatial distribution of (a) MODIS Aqua AOD (550 nm) daily product MYD04 10 km; (b) AOD (550 nm) from PLASMA sun photometer on-road measurements along Lille-Dunkerque transect on 26 August 2016 are overlapped on MYD04 AOD product.





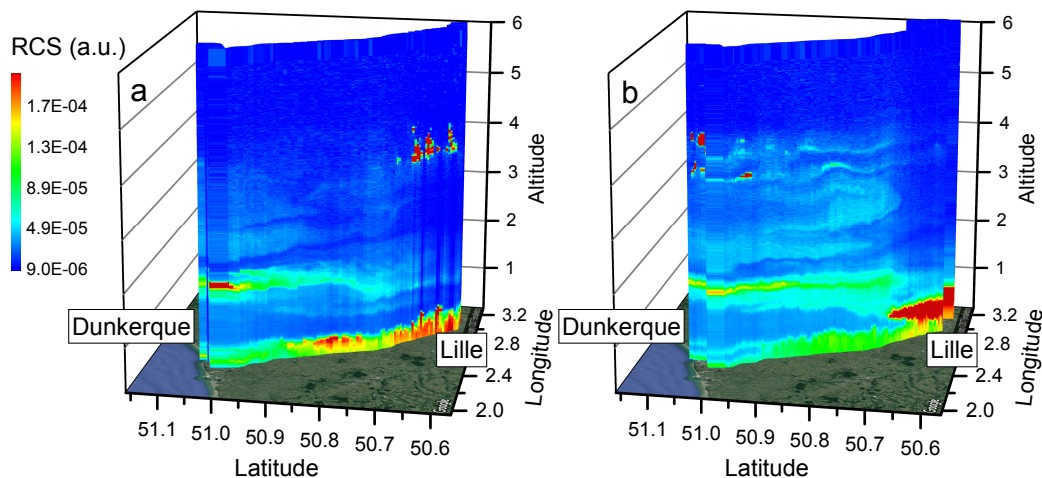

**Figure 6. Three-dimensional view on the spatio-temporal variability of the LiDAR range-corrected signals measured along (a) Lille-Dunkerque and (b) Dunkerque-Lille transects on 26 August 2016. Observed structures: PBL up to 1 km altitude, elevated aerosol layers between 2 and 5 km and scattered clouds at 4 km near Lille and Dunkerque represented with red color.**

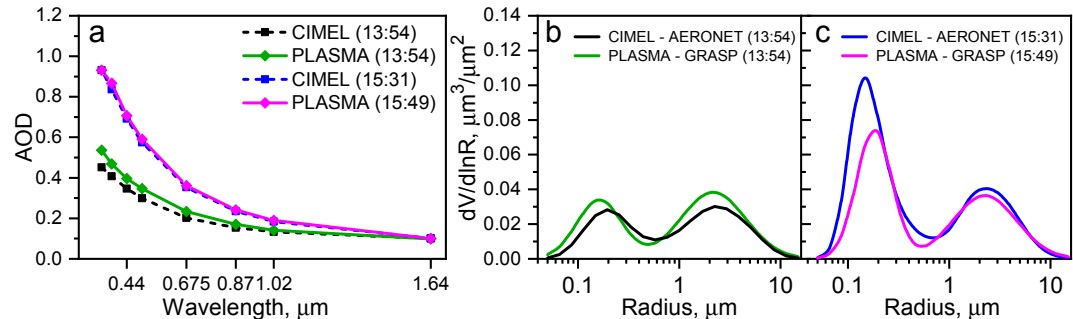

**Figure 7. (a) Spectral AOD from PLASMA sun photometer measurements at Dunkerque, on 26 August 2016, 13:54 UTC (green solid line) and Lille, 15:49 UTC (magenta solid line). The closest spectral AOD from Dunkerque and Lille AERONET sites are represented with dashed lines. The corresponding total column aerosol volume size distribution retrieved by GRASP-AOD inversions for (b) Dunkerque and (c) Lille sites. The size distributions from the closest AERONET standard inversion for Dunkerque (13:54) and Lille (15:31) are also represented as reference.**



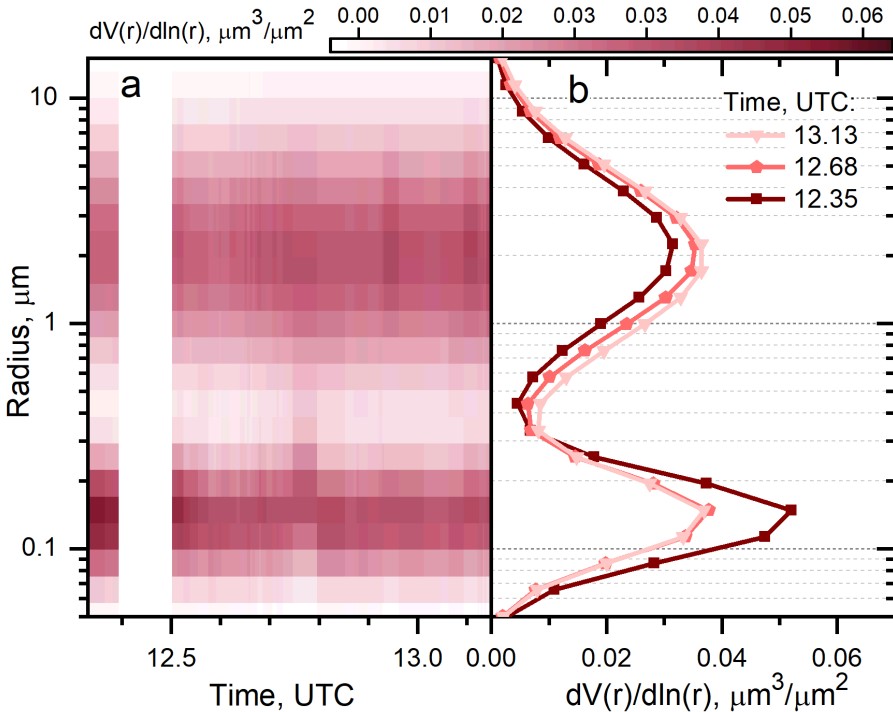

**Figure 8.** Spatio-temporal variability of total column aerosol volume size distribution retrieved with GRASP-AOD using mobile sun photometer measurements along Lille-Dunkerque transect on 26 August 2016 (12:18-13:10 UTC).





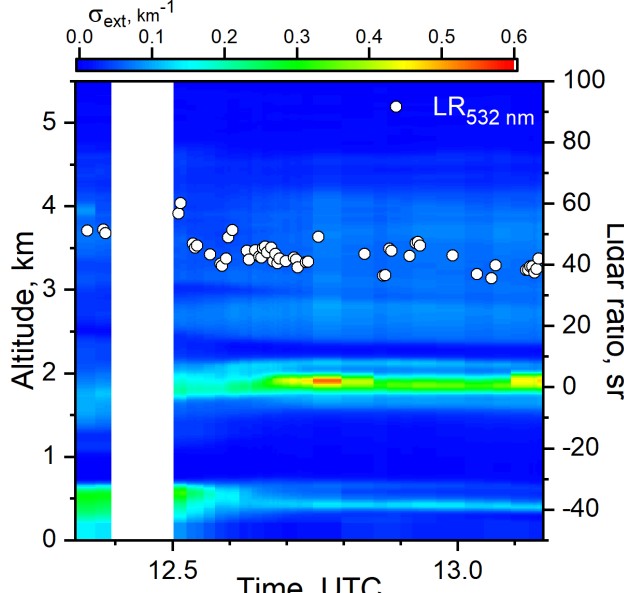

**Figure 9. Spatio-temporal variability of extinction coefficient (colormap) and extinction-to-backscatter ratio (white dots) at 532 nm from on-road mobile measurements along Lille-Dunkerque transect on 26 August 2016 (12:18-13:10 UTC).**





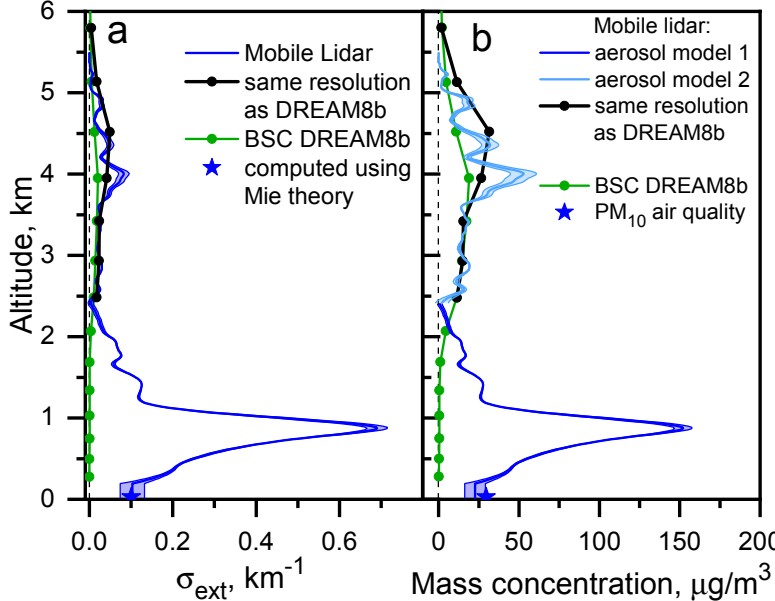

**Figure 10. Comparison of (a) extinction coefficient and (b) mass concentration mean profile at Lille, 15:30 UTC from mobile lidar-sun photometer measurements (532 nm) (blue) with BSC-DREAM8b mean dust extinction and concentration at 550 nm (green). Data points from mobile measurements in the range 2.5-5 km are reduced to the model resolution (black) for comparison with the simulated dust profiles.**





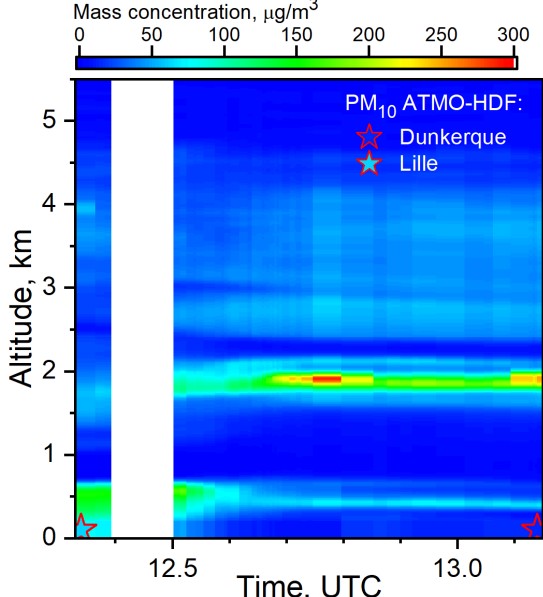

**Figure 11.** Spatio-temporal variability of lidar-sun photometer-derived mass concentration (colormap) from on-road measurements along Lille-Dunkerque transect on 26 August 2016 (12:18-13:10 UTC). The PM$_{10}$ mass concentrations from closest measurements from air quality stations along the route are overlapped (colormapped stars).



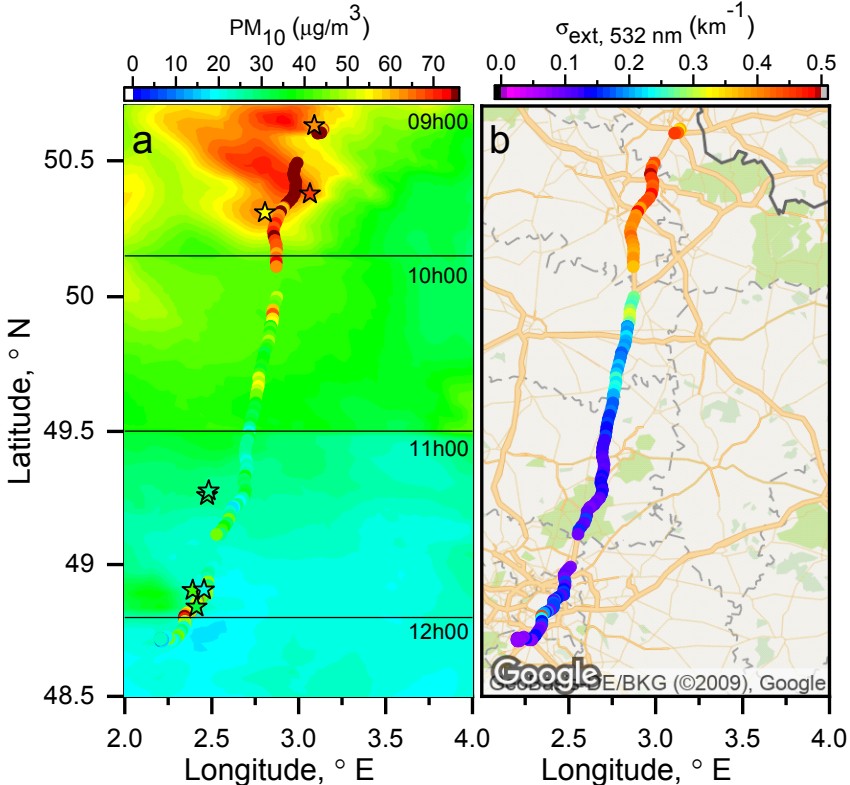

**Figure 12.** Spatial variability of (a) modelled PM₁₀ concentrations and PM₁₀ measured by mini-WRAS during mobile observations along Lille-Palaiseau transect on 28 August 2017. Air quality stations along the road are marked with colormapped stars. The colormap is the same for both the measured and the modelled PM₁₀ concentrations . (b) extinction coefficient at 532 nm at ground level computed using Mie calculations on aerosol spectrometer data.