# Peer review of "Description and applications of a mobile system performing on-road aerosol remote sensing and in situ measurements"

_Atmospheric Measurement Techniques, 2018_

## Referee Comment (RC1) · Anonymous Referee #1 · 26 Apr 2018

Much progress has been maid on ground networks of surface PM2.5, aerosol optical depth and lidar, however, the spatial coverage of such surface networks is still limited and these networks cannot reveal subtle spatial variation of these key aerosol parameters. Mobile facility equipped with sophisticated instruments capable for simultaneous measurements of aerosol properties is a good idea to fill the gap left by the surface networks. A mobile platform instrumented by sunphotometer, lidar and OPC was described in detail in this manuscript. The manuscript provides a detailed technical documentation of this platform. Some interesting results were also presented including validation of satellite remote sensing and model aerosol products. Overall, this is a nice manuscript and I suggest to accept it after a minor revision.

[Figure]

1. The structure of the paper is a little complex and it looks somewhat a technical report. I suggest to present the description of the system according to its instruments including its technical detail, its data processing procedure, as well as its uncertainty. For example, in section of methodology, texts related to the introduction of PLASMA can be combined together and therefore, it is easy for readers to have a good understanding of this instrument and application in the mobile platform.

2. I suggest to add a discussion section in which to talk about the status of this mobile facility and its potential developments in near future, also can talk about its potential application in atmospheric environment study and climate research, therefore, these sentences in section 2 and other sections can be combined together to present a clear picture about the status, its uncertainty, its potential developments, as well as its applications.

3. Grimm was said to can work under condition when air speed is 25 m/s, i.e., 90 km/hr, this means that the car speed should be within this threshold, otherwise, in situ measurements would be impacted. Additionally, Grimm measurement of size distribution is based on both electrical mobility and optical method.

Minor issue: 1.page 4, fulfils to fulfills 2. suggest to change methodology to instrument, measurement and data quality control, more specific 3. page 9, how to control of data quality in the presence of tree and building

---

## Short Comment (SC1) · 11 May 2018

The authors describe a mobile system equipped with a micropulse lidar, a sun photometer and an aerosol spectrometer, and its abilities for performing on-road measurements to derive aerosol properties including aerosol optical depth, volume size distribution, extinction coefficient profiles, extinction-to-backscatter ratio, particle number, and mass concentration. This intensive platform integrated with both remote sensing and in situ instruments shows useful in comprehensive study of aerosols, as well as validations of satellite products and model forecasts. The manuscript is globally well-written. But I advise the authors to compress the paper. Moreover, I still expect to see

some results with joint application of the active and passive remote sensing, and in-situ instruments. Such as, comprehensive analysis of multi-instrument observations, or the possibilities of constrain and joint inversion by different instruments. I hence recommend to consider these aspects before publication.

A few more detailed comments: 1. Line 2, page 10: The GRASP algorithm and software was used to retrieve the volume size distribution from the PLASMA sun photometer measurements. Why don't you use the GRASP to do joint retrieval of aerosol properties from the PLASMA sun photometer and Lidar? 2. Line 19, page 12: Is there any influence of the fluctuations of platform on the uncertainty of AOD from PLASMA sun photometer? 3. Line 25, page 14, The difference between PLASMA and MODIS AOD results is obvious, especially in high aerosol loading areas around Lille (see Figure 5). It should be discussed. 4. Line 15-17, page 19: Please give more detail on the uncertainty estimation.

---

## Author Comment (AC1) · 3 Jul 2018

We would like to thank the two reviewers for their careful reading and the time dedicated, that are highly appreciated. The general remarks provided as well as the specific comments are an important feedback, which help us improve the scientific content and the manuscript's quality. Please find below our point-to-point replies. The referees' comments (RCs) are listed below in black and the authors' answers are listed below in blue.

**Anonymous Referee #1**

Much progress has been maid on ground networks of surface PM2.5, aerosol optical depth and lidar, however, the spatial coverage of such surface networks is still limited and these networks cannot reveal subtle spatial variation of these key aerosol parameters. Mobile facility equipped with sophisticated instruments capable for simultaneous measurements of aerosol properties is a good idea to fill the gap left by the surface networks. A mobile platform instrumented by sunphotometer, lidar and OPC was described in detail in this manuscript. The manuscript provides a detailed technical documentation of this platform. Some interesting results were also presented including validation of satellite remote sensing and model aerosol products. Overall, this is a nice manuscript and I suggest to accept it after a minor revision

1. The structure of the paper is a little complex and it looks somewhat a technical report. I suggest to present the description of the system according to its instruments including its technical detail, its data processing procedure, as well as its uncertainty. For example, in section of methodology, texts related to the introduction of PLASMA can be combined together and therefore, it is easy for readers to have a good understanding of this instrument and application in the mobile platform.

We have combined section 2 and section 3 in one section called: "Instruments, methodology and data quality", as you suggested. We thank you very much for this suggestion and we hope this will make it clearer for the readers.

2. I suggest to add a discussion section in which to talk about the status of this mobile facility and its potential developments in near future, also can talk about its potential application in atmospheric environment study and climate research, therefore, these sentences in section 2 and other sections can be combined together to present a clear picture about the status, its uncertainty, its potential developments, as well as its applications.

We have added a discussion section (section 4), where the status, applications and potential developments of the mobile system are summarized. We thank the reviewer for this suggestion. It really helps better structure the manuscript.

3. Grimm was said to can work under condition when air speed is 25 m/s, i.e., 90 km/hr, this means that the car speed should be within this threshold, otherwise, in situ measurements would be impacted. Additionally, Grimm measurement of size distribution is based on both electrical mobility and optical method.

We have performed tests to observe the impact of the vehicle's driving speed onto the aerosol spectrometer measurements. We did not observe any influence of speeds exceeding 90 km/hour on the measurements. Please find below a figure illustrating the variability of PMx recorded on roads and the vehicle's speed variation. The horizontal green lines in the bottom figure represent the minimum and maximum limits of air velocity as defined by the manufacturer for the 1 mm diameter inlet used for the isokinetic probe for mobile sampling. We could expect a different behaviour, e.g. increase of PMx concentrations or erroneous values of PMx, when the speed exceeded 90 km/h, which was not observed during our measurements. Furthermore, we checked if the recorded values during vehicles' motion were realistic and comparable to the values from regional air quality stations closest to the route. The measurements were consistent and within the range of values recorded at the stations. These tests showed that sampling at a speed of 110 km/h would not affect the measurements. For reasons of time efficiency (especially for the long transects) and taking into account that mostly, the mobile measurements were performed along motorways, a higher speed was chosen for mobile observations.

[Figure]

Figure 1 R1. Example of temporal variability of $PM_{10}$, $PM_{2.5}$ and $PM_1$ mass concentrations derived from GRIMM mini-WRAS mobile measurements (upper figure) and vehicle's speed (bottom figure).

Minor issue:

1.page 4, fulfils to fulfills

Corrected, thank you.

2. suggest to change methodology to instrument, measurement and data quality control, more specific

We have combined section 2 and section 3 in one section called: "Instruments, methodology and data quality control", as you suggested. We thank you very much for this suggestion and we hope this will make it clearer for the readers.

3. page 9, how to control of data quality in the presence of tree and building

This has been explained in the original manuscript at page 5, lines 24-26 and page 9, lines 5-7. However, we reformulated the explanation of sun photometer data processing as to make it clearer and we have inserted it in the corrected manuscript. Please find the answer below:

*The sun photometer measurements contaminated by obstacles along the transect such as*

*clouds, buildings, trees, bridges, etc., are filtered using the triplet stability criterion described by Smirnov et al. (2000). The filtering is applied on the recorded digital signals and consists in applying a threshold of 1% to 3% maximum difference between three consecutive measurements within a defined time window (from 10 seconds to 30 seconds for stationary measurements). The threshold value relates to the expected AOD variability in a stable atmosphere within the defined time window and is chosen by the user at the time of processing the data. If the condition is not met at any wavelength, the measurements at all wavelengths are eliminated from further processing.*

**Z. Li (Referee) #2**

The authors describe a mobile system equipped with a micropulse lidar, a sun photometer and an aerosol spectrometer, and its abilities for performing on-road measurements to derive aerosol properties including aerosol optical depth, volume size distribution, extinction coefficient profiles, extinction-to-backscatter ratio, particle number, and mass concentration. This intensive platform integrated with both remote sensing and in situ instruments shows useful in comprehensive study of aerosols, as well as validations of satellite products and model forecasts. The manuscript is globally well-written. But I advise the authors to compress the paper. Moreover, I still expect to see some results with joint application of the active and passive remote sensing, and in- situ instruments. Such as, comprehensive analysis of multi-instrument observations, or the possibilities of constrain and joint inversion by different instruments. I hence recommend to consider these aspects before publication.

We thank the reviewer for his suggestions. We re-organized the manuscript as to make it clearer. We compressed the sections 2 and 3 in one section called "Instruments, methodology and data quality control" and we added a section "Discussion" that summarizes the status, applications and future developments of the mobile system. We hope these changes will give a clearer structure to the manuscript.

Regarding the joint application of remote sensing and in situ measurements we added in the manuscript a sub-section illustrating this application. We must though mention that illustrating this application of joint in situ and remote sensing inversion was not possible for the case studies discussed in the manuscript. For the first case of 26[th] August 2016 the aerosol spectrometer was not integrated in the mobile system at that time and for the second case of 28[th] August 2018, presence of clouds made impossible the joint retrieval of remote sensing observations. For these reasons we chose to illustrate the joint inversion of in situ and remote sensing observations on a different case, 7[th] July 2017 in France, on a transect on the South to North axis.

A few more detailed comments:

1. Line 2, page 10: The GRASP algorithm and software was used to retrieve the volume size distribution from the PLASMA sun photometer measurements. Why don't you use the GRASP to do joint retrieval of aerosol properties from the PLASMA sun photometer and Lidar?

For practical and historical reasons we have chosen to show retrievals with BASIC

(Mortier, 2013) as the routines for data preparation and processing have been already done and the results can be obtained faster. We considered that for the concept of mobile system and automatic processing and retrievals in near-real time, BASIC retrievals are more suited to be shown at this stage of development of the mobile system. Regarding GRASP/GARRLiC (Lopatin et al., 2013) retrievals, the routines for preparing the data in the right input format has been achieved recently and can be used in the future for automatic processing. Please find below a figure with an example of extinction coefficient profiles retrieved with both BASIC and GRASP/GARRLiC algorithm. One can see that the results are similar. For the final manuscript we have chosen to show the results obtained with BASIC.

[Figure]

Figure 2 R2. Example of BASIC and GRASP/GARRLiC retrievals for 26th August 2016, 12 :30 UTC, France.

2. Line 19, page 12: Is there any influence of the fluctuations of platform on the uncertainty of AOD from PLASMA sun photometer?

As it is explained in the initial manuscript at page 5, lines 24-26, we use the triplet stability criterion to filter bad data that could come from encountering obstacles along the roads such as buildings and trees and also to eliminate unstable measurements. If fluctuations of the platform would impact the measurements, this would be seen as high signal variations at small time steps. This would be eliminated by our filter that considers three consecutive measurements in a time window from 10 seconds to 20 seconds.

We reformulated the explanation of sun photometer data processing as to make it clearer

and we have inserted it in the corrected manuscript. Please find the answer below:

*The sun photometer measurements contaminated by obstacles along the transect such as clouds, buildings, trees, bridges, etc., are filtered using the triplet stability criterion described by Smirnov et al. (2000). The filtering is applied on the recorded digital signals and consists in applying a threshold of 1% to 3% maximum difference between three consecutive measurements within a defined time window (from 10 seconds to 30 seconds for stationary measurements). The threshold value relates to the expected AOD variability in a stable atmosphere within the defined time window and is chosen by the user at the time of processing the data. If the condition is not met at any wavelength, the measurements at all wavelengths are eliminated from further processing.*

3. Line 25, page 14, The difference between PLASMA and MODIS AOD results is obvious, especially in high aerosol loading areas around Lille (see Figure 5). It should be discussed.

MODIS AOD map represents a "snapshot" of the atmosphere's integrated aerosol content at the time of Aqua satellite overpass over the region (12:05 UTC), while the mobile measurements overlapped on the map correspond to the time interval 11:53-13:16 UTC. We started our mobile transect from Villeneuve d'Ascq (a suburb of Lille) at 11:53 UTC. A possible reason for the difference at Lille could be that at the Aqua overpass time, a higher aerosol concentration and/or presence of clouds could have been recorded in the 10 km area. Regarding the rest of the transect, the differences are harder to quantify since the measurements' time of MODIS and PLASMA are different. Nevertheless, a mean absolute difference of 0.14 was obtained over the rest of the transect, which is acceptable taking into account all the variables that can explain this difference (i.e. different temporal and spatial resolution, accuracy for both instruments and products, atmospheric variability within 1 hour of measurements). We consider that the results of this first comparison are encouraging, taking into account the above mentioned parameters, and shows the importance of the pursuit of such studies for evaluating satellite products.

4. Line 15-17, page 19: Please give more detail on the uncertainty estimation.

The uncertainties on the measurements and on the retrievals were discussed in sub-section 3.2.4 and are now re-organized in the sections presenting details for each instrument. For this analysis we considered the uncertainties estimated by Mortier et al. (2013).

**References :**

Lopatin,  a., Dubovik, O., Chaikovsky, A., Goloub, P., Lapyonok, T., Tanré, D. and Litvinov, P.: Enhancement of aerosol characterization using synergy of lidar and sun-photometer coincident observations: The GARRLiC algorithm, Atmos. Meas. Tech., 6(8), 2065–2088, doi:10.5194/amt-6-2065-2013, 2013.

Mortier, A.: Tendances et variabilites de l'aerosol atmospherique a l'aide du couplage Lidar/Photometre sur les sites de Lille et Dakar, University of Lille., 2013.

Mortier, A., Goloub, P., Podvin, T., Deroo, C., Chaikovsky, A., Ajtai, N., Blarel, L., Tanre, D. and Derimian, Y.: Detection and characterization of volcanic ash plumes over Lille during the Eyjafjallajökull eruption, Atmos. Chem. Phys., 13(7), 3705–3720, doi:10.5194/acp-13-3705-2013, 2013.

---

## Author Comment (AC2) · 3 Jul 2018

Dear Sir/Madam,

Please find attached in the supplement to this comment the answers to the reviewers' questions and comments. We thank the reviewers for their constructive comments that we believe will give a better structure to the paper. All coauthors have contributed to these answers and agree with the submission.

Yours sincerely,

Ioana Elisabeta Popovici

[Figure]

Please also note the supplement to this comment:
https://www.atmos-meas-tech-discuss.net/amt-2018-103/amt-2018-103-AC2-
supplement.pdf

———————————————————